# Evolution of sexual systems, sex chromosomes and sex-linked gene transcription in flatworms and roundworms

Yifeng Wang [1], Robin B. Gasser [2], Deborah Charlesworth[3✉] & Qi Zhou [1,4,5,6✉]

Many species with separate male and female individuals (termed 'gonochorism' in animals) have sex-linked genome regions. Here, we investigate evolutionary changes when genome regions become completely sex-linked, by analyses of multiple species of flatworms (Platyhelminthes; among which schistosomes recently evolved gonochorism from ancestral hermaphroditism), and roundworms (Nematoda) which have undergone independent translocations of different autosomes. Although neither the evolution of gonochorism nor translocations fusing ancestrally autosomal regions to sex chromosomes causes inevitable loss of recombination, we document that formerly recombining regions show genomic signatures of recombination suppression in both taxa, and become strongly genetically degenerated, with a loss of most genes. Comparisons with hermaphroditic flatworm transcriptomes show masculinisation and some defeminisation in schistosome gonad gene expression. We also find evidence that evolution of sex-linkage in nematodes is accompanied by transcriptional changes and dosage compensation. Our analyses also identify sex-linked genes that could assist future research aimed at controlling some of these important parasites.

[1] MOE Laboratory of Biosystems Homeostasis & Protection and Zhejiang Provincial Key Laboratory for Cancer Molecular Cell Biology, Life Sciences Institute, Zhejiang University, Hangzhou 310058, China. [2] Department of Veterinary Biosciences, Melbourne Veterinary School, The University of Melbourne, Parkville, Victoria 3010, Australia. [3] Institute of Evolutionary Biology, School of Biological Sciences, University of Edinburgh, West Mains Road, Edinburgh EH9 3LF, UK. [4] Evolutionary & Organismal Biology Research Center, School of Medicine, Zhejiang University, Hangzhou 310058, China. [5] Department of Neuroscience and Developmental Biology, University of Vienna, Vienna 1090, Austria. [6] Center for Reproductive Medicine, 2nd Affiliated Hospital, School of Medicine, Zhejiang University, Hangzhou 310058, China. ✉email: Deborah.Charlesworth@ed.ac.uk; zhouqi1982@zju.edu.cn

In eukaryotes, many independent evolutionary transitions have been documented between hermaphroditic sexual systems and separate sexes (which is termed dioecy in plants and gonochorism in animals)[1]. Dioecious plants or gonochoristic animals are characterised by individuals of the same species but with distinct male or female reproductive organs and usually other sexually dimorphic traits. While gynodioecy (polymorphism of females and hermaphrodites) is regularly observed in plants, and may sometimes represent an intermediate step in the evolution of dioecy[2–7]), it is very rare in animals, with only one verified gynodioecious animal species reported so far[8].

Androdioecy (polymorphism of males and hermaphrodites) is rare in both plants and animals, and generally represents a transition from gonochorism to hermaphroditism (as in the androdioecious *Caenorhabditis elegans*), likely due to selection for reproductive assurance rather than the reverse direction of change[2,9–11]. In animals, transitions from hermaphroditic ancestors to androdioecy may be associated with selection for sexual dimorphism in size[10]. For example, almost all flatworms are exclusively hermaphroditic, except for parasitic species of the family Schistosomatidae (blood flukes). Female schistosomes are much smaller than males, which could be adaptive for 'precision egg' laying in hosts' vascular systems[5,6], and they cannot develop mature reproductive organs without pairing with males[6]. The molecular mechanisms and genomic consequences of evolutionary transitions between different sexual systems, and the effects on genes controlling sexual dimorphism, are rarely known.

The *de novo* evolution of separate sexes has mostly been studied in flowering plants[4,12] rather than in animals. Plants are predominantly hermaphroditic, and angiosperms are estimated to have independently evolved dioecy hundreds of times[13,14]. However, excluding insects, around one-third of animal species are hermaphroditic[15], and transitions to gonochorism have occurred within flatworms, annelids, and mollusks[1]. Evolution of separate sexes from hermaphroditism requires at least two mutations, an initial mutation that produces females, followed by a second mutation to produce males (or vice versa). Selection then favours closer linkage between the two loci, potentially leading to a fully sex-linked region that may include other genes without sex-determining functions[2]. Genetic evidence supporting the involvement of two genes has been found in plants including *Silene latifolia*, asparagus and kiwifruit[4,12]. Alternatively, the two mutations can lead to dioecy with a single sex determining gene ('one-factor' systems) that controls male or female development, as suggested by the cases of persimmon and *Populus*[2,16]. However, the single master sex-determining genes reported in most animals, including mammals and many teleosts, probably originated after, not with, the transition to gonochorism; this involved replacing an earlier sex-determining gene in a so-called 'turnover event'[17,18]. Recombination suppression in these species is hypothesised to have evolved to reduce crossovers between the sex-determining locus and sexually antagonistic polymorphic alleles at a closely linked locus[19]. Separate recombination suppression events were inferred to have occurred at different times in the history of species including schistosomes[20], sticklebacks[21,22], birds[23] and mammals[24], producing stratified sequence divergence levels between different sex-linked regions, termed 'evolutionary strata'[25,26].

Whether recombination suppression evolved with or after the origination of separate sexes, fully Y- or W-chromosome linked regions are expected to diverge from their counterparts (X or Z, respectively), and given enough time, undergo genetic degeneration, and a loss of genes[27]. This consequence of recombination suppression has been demonstrated by studies of the ancient and degenerated mammalian Y and avian W chromosomes, and of degenerating 'neo-sex' chromosomes created by recent fusions between ancestral sex chromosomes and autosomes, which we outline next[28].

In dipteran flies[29] and lepidopterans[30], chromosomal crossovers and thus recombination occur only in one sex, with male flies and female lepidopterans having achiasmate meiosis. A formerly autosomal neo-sex region in these species therefore immediately stops recombining after a fusion with an ancestral sex chromosome. In contrast, in species that recombine in both sexes, recombination is expected to initially continue on a former autosome after fusing to a sex chromosome. Interestingly, however, neo-sex chromosomes of some such species, including mammals[24] and sticklebacks[22], have subsequently evolved non-recombining evolutionary strata. It remains unclear whether, why, and how recombination has become suppressed on the neo-sex chromosomes of species that have crossovers in both sexes, though chromosome inversions are probably sometimes involved.

Here we investigate representative species of two speciose and widely distributed phyla, the Platyhelminthes (flatworms including the classes Trematoda and Cestoda) and the Nematoda (Nemathelminthes, roundworms or nematodes), to reconstruct the origin and evolution of sex chromosomes. These phyla diverged about 700 million years ago (MYA)[31]. As mentioned above, flatworms are almost exclusively hermaphroditic[5,6], but the schistosomes have evolved gonochorism and sex chromosomes de novo. This group of species is therefore particularly interesting for studying the transition from hermaphroditism to gonochorism. Here we use *Schistosoma mansoni* and its hermaphroditic outgroups to study evolutionary strata and changes in gene transcription in response to this transition.

In contrast, the vast majority of nematodes are gonochoristic, and the reverse transition, breakdown of gonochorism to secondary hermaphroditism[32], has occurred in some free-living species, including the androdioecious *C. elegans* and *C. briggsae* (Fig. 1). Genome sequencing of nematode species including *Onchocerca volvulus*[33] and *Brugia malayi*[34] indicated that their sex chromosomes include different ancestral linkage groups, or 'Nigon elements'[35,36] (the nematode analogs of the Muller elements in *Drosophila*[37]). These seven chromosome elements were first proposed after comparing the genomes of rhabditid nematodes (order Rhabditida, including clade III, clade IV and clade V, see below)[36,38], most of which have a karyotype with $n = 6$. Orthologous genes are found on homologous chromosomes across different species, defining the Nigon elements, although gene orders within elements differ because of rearrangements, and different elements may undergo fusions or fissions. The five *C. elegans* autosome pairs correspond to elements NA to NE, and individual orthologous genes are rarely found on different elements. However, the NN and NX elements that, together, constitute the *C. elegans* X (see also below) are exceptions. We now study the sex chromosomes of species from all major nematode clades, and assign their genes to the component Nigon elements, based on orthology with *C. elegans* genes. We also identify sex-linked genes in many species, including previously undetected Y- and W-linked genes, and finally characterise their transcription profiles in the two sexes, allowing us to reconstruct the likely chromosomal ancestors of their sex chromosomes, and ask whether sex chromosome evolution is accompanied by changes in patterns of recombination and/or gene transcription.

Overall, we describe the time-course of evolution of differences within sex-linked regions after they evolved in flatworms and roundworms, including evidence concerning suppressed recombination, the evolution of heteromorphism and genetic degeneration, and the associated evolution of dosage compensation. Our results provide a preliminary foundation for the future discovery of the sex-determining genes of these species, which could potentially lead to the ability to interfere with or disrupt

reproductive and developmental processes in these parasitic worms that cause diseases in human and other animals[39,40].

## Results

### Identifying sex-linked regions in schistosomes and nematodes.
To systematically identify the sex-linked regions in each of these two deeply diverged phyla, we compiled sexual system information as well as published genomic (references shown in Supplementary Data 1) and transcriptomic (Supplementary Data 2) data from sexed individuals (when available) for 13 platyhelminths and 41 nematodes species (Fig. 1a). Genome sequencing quality varied markedly among these published datasets (Supplementary Data 1, with references of all used data included). For 11 nematodes and 3 platyhelminth species, chromosome-level genome assemblies are available, and 4 nematode chromosomal genome assemblies (from *Trichuris muris* in clade I, *Brugia malayi* in clade III, *Strongyloides ratti* in clade IV and *C. elegans* in clade V) were used as references to create chromosome assemblies from scaffold-level draft genomes of 19 other related species (Supplementary Data 3; and between 60% and 99% of the sequences can be incorporated into the assemblies). Previous cytogenetic studies suggested a ZZ/ZW sex system in the ancestor of schistosomes[41], and an XX/XO system in the ancestor of nematodes[42,43]. If extensive Y or W-linked regions of species in either phylum have become highly degenerated due to lack of recombination[27,28], the X or Z-linked regions can be identified from low genomic read coverage in the heterogametic sex (females in schistosomes, and males in nematodes). Such low coverage results from substitutions accumulated in Y- or W-linked sequences, together with Y- or W-specific transposable element insertions, both of which hinder mapping of sequencing reads to their counterpart X- or Z-linked regions, plus the direct effect on coverage of large deletions[28]. Coverage analysis, without linkage map data for the sex chromosomes, can thus detect highly divergent sex-linked regions[44], and is widely used, including in birds[23], teleosts[22], reptiles and the Lepidoptera[45]. It has also previously been used to detect sex-linked regions in schistosomes[20], and to identify the X-linked regions in several nematode species[34,46]. Our analysis looked for bimodal distributions of male or female read coverage, with separate peaks corresponding to autosomal and sex-linked regions. For all 8 species whose sex chromosomes have previously been identified, our coverage pipeline yielded results in agreement with the published ones[33,34,46–51], confirming that this approach is reliable (Supplementary Data 1). Physically small or non-differentiated sex-linked regions, however, may not be detectable by our approach.

Among all the platyhelminth species studied, only the schistosome genome sequences exhibited the signature indicating presence of sex chromosomes, and (as expected since males are ZZ) it was not seen when only male reads were available (Fig. 1b; Supplementary Data 4). Hermaphroditic trematodes and cestodes also exhibited similar coverage for all chromosomes or a unimodal coverage distribution, again as expected. By comparing

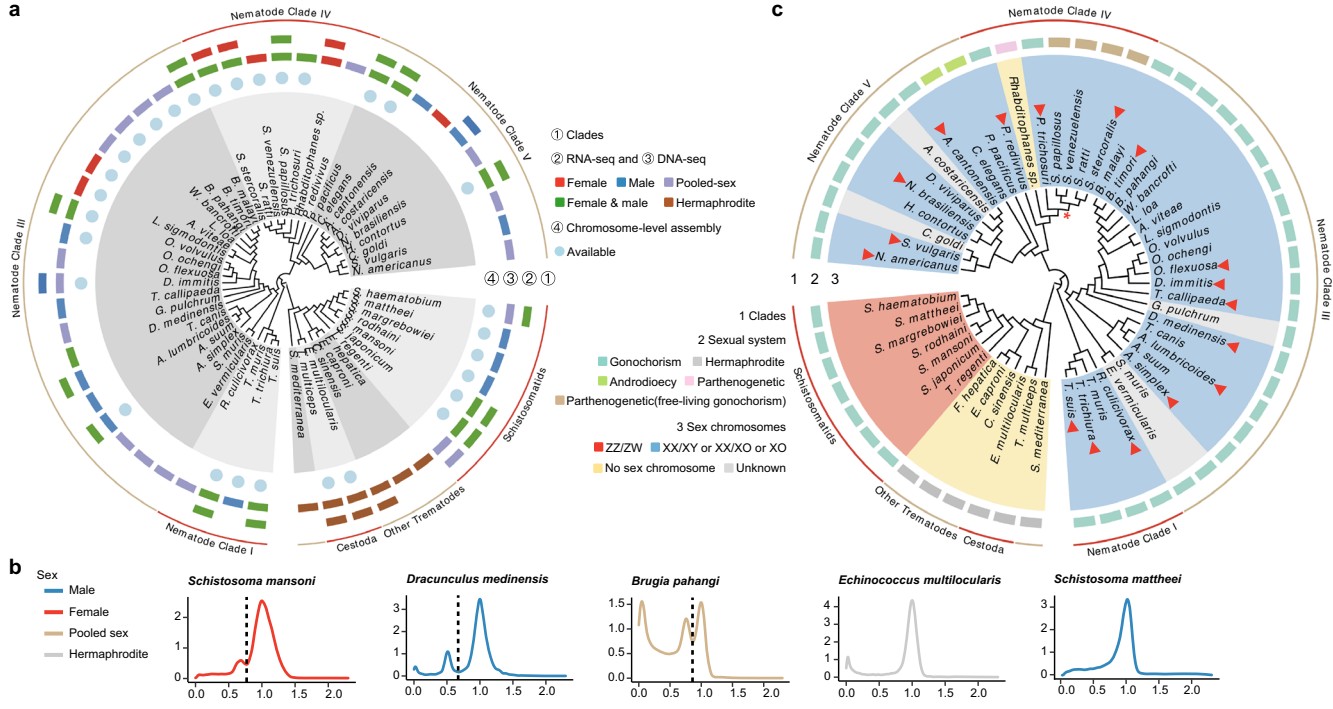

**Fig. 1 Sex chromosomes of platyhelminths and nematodes. a** Genomic and transcriptomic data used in this study. We obtained available genomic and transcriptomic sequences for a total of 13 platyhelminths species and 41 nematode species, along with information about the sex of each sample. **b** Examples of read coverage distributions of the species showing bimodality allowing us to identify the completely sex-linked sequences, or unimodality (in which we cannot identify the sex-linked sequences). The sexes are indicated by different line colours. **c** Sex chromosomes and sexual systems of the species studied. The outer to the inner rings show taxonomic information, and the sex and sex chromosome systems. Red arrowheads indicate the species for which we newly identified sex-linked sequences or newly annotated Nigon elements of their sex chromosomes in this study. For some species for which only data for the homogametic sex were available (including *S. mattheei* and *S. margrebowiei*), we labeled their sex chromosome type based on previously reported cytogenetic evidence (see Supplementary Data 4). We marked the parasitic *Strongyloides* clade here and in Fig. 2d below with an asterisk to indicate that this group (including *S. ratti* and *S. papillosus*) have environmental sex determination. Males of this group develop in response to the host immune responses, which triggers loss of X-linked sequences during oocyte mitosis[55]. EvolView version 2 (http://www.evolgenius.info/evolview/)[117] was used to display the phylogenetic tree. Source data are provided as a Source data file.

the previously identified 'oldest' evolutionary stratum (stratum 0 or S0) of *Schistosoma japonicum*[52] to the sex-linked region of the distantly related *S. haematobium* (Supplementary Fig. 1), we confirmed the previous conclusion that they share the same S0 region. Therefore, recombination was first suppressed between the Z and W chromosomes in the ancestor of all schistosomes[20], estimated to be about 21 MYA[53], and the younger strata evolved subsequently. Our results also support previous data[5,20] suggesting that gonochorism and the presence of sex-linked degenerated regions have originated only once in flatworms, with only a single ancestral linkage group having evolved to become a sex chromosome pair.

In the nematodes, different chromosomes became sex-linked in different clades. Our analyses newly identified the sex-linked regions and/or allowed the annotation of their corresponding Nigon elements in 17 species (indicated by red arrowheads in Fig. 1c). This was possible for almost all of the species with the expected bimodal read coverage distributions in males (Fig. 1b, Supplementary Figs. 2–4), whereas most parthenogenetic species did not show bimodal read coverage (see below), or species with reads available only in females, or with both sexes pooled (Fig. 1b, c). In addition to identifying sex-linked sequences in the homogametic sex of the species studied, we also sought to identify Y- or W-linked genes. As such genes may not be well assembled, due to the presence of repetitive sequences or very long introns, we further identified genes whose de novo assembled transcriptome sequences showed sex differences in genomic read coverage values from sexed animals (Supplementary Fig. 5). Combined with genes directly assembled from genomic reads, we annotated between 6 and 219 Y-linked genes in 17 individual nematode species studied, and 48 W-linked genes for *S. mansoni* (Fig. 1c, Supplementary Data 4).

**Recurrent chromosome translocations involving nematode sex chromosomes.** Previous studies reported translocations between sex chromosomes and autosomes in some nematode species[34,36,38]. Our findings confirm these results, but also identify new translocations, and we now characterise the order and timing of these rearrangements in relation to the phylogenetic relationships, demonstrating independent events over a broad evolutionary time scale (Fig. 2).

We first focused on seven representative species with chromosome-level genome assemblies available (six from the order Rhabditida, i.e., clade III, clade IV and clade V species, one from clade I species) (Supplementary Data 1), and compared their X-linked gene contents and genome sequences. The autosomes of *C. elegans* (clade V species) correspond to Nigon elements NA to NE, and its X chromosome corresponds to two elements, NX and NN (X + N)[36]. The X + N composition is based on alignments of *C. elegans* X-linked genes to those in the genomes of five other Rhabditida nematodes: 103 *C. elegans* genes' orthologs are also located on the X chromosomes of all five species, and are classified as NX-derived genes (Fig. 2a, b; Supplementary Fig. 6). The orthologs of the other 417 genes are classified as the NN-derived genes. These orthologs are either autosomal (in another clade V species - *Pristionchus pacificus*, in clade IV species - *Strongyloides ratti*, and clade III species - *O. volvulus*; Fig. 2a; Supplementary Fig. 7), or are found on the X chromosome, but in a distinct region from that occupied by the NX-derived ones (in the clade V species - *Haemonchus contortus*, and the clade III species - *B. malayi*). These findings suggest that the NX element was already a sex chromosome in the ancestor of rhabditid species 241 MYA (http://www.timetree.org/) or even earlier (see Fig. 2c below), and that sex chromosome-autosome fusions or translocations added the NN element independently in some clade III and clade IV

species (Fig. 2a). The sex chromosomes of the Rhabditida thus originated much earlier than the origin of the oldest sex-linked region in extant schistosomes.

The only non-rhabditid species in Fig. 2, *Trichuris muris* from clade I, has a sex chromosome composition of NA and NB elements (A + B), and lacks genes derived from the NX element. The divergence between clade I and the Rhabditida is dated at 400 million years ago[54]. During this time, either separate sex chromosomes originated independently in clade I and the ancestor of the Rhabditida, or an ancestral NX element was replaced in a turnover event in clade I, with either the NA or NB element subsequently becoming the sex chromosome, followed by a fusion with the other element, creating a neo-sex chromosome (see below).

Based on seven species with chromosomal genome assemblies shown in Fig. 2, we infer that, in the non-clade I or rhabditid species, the NX element has undergone independent translocations involving other elements, repeatedly forming neo-sex chromosomes. In clade V species, the NN element has translocated to the NX element, forming an X + N sex chromosome in *C. elegans* and *H. contortus*, but not in the *P. pacificus* lineage. In clade III, the NE and ND elements have translocated to the NX in *O. volvulus* (X + E + D)[33], and the NN and ND elements have translocated to the NX in *B. malayi* (X + N + D)[34] (Fig. 2c). As the X + D translocation was found in species within clade III and clade IV, but not in the clade V species studied, it could have occurred either in the common ancestor of the Rhabditida, or independently in the ancestors of the respective clades. The former scenario seems unlikely, as it requires an X + D translocation followed by a complete fission and reversion of the ND element to a separate autosome in the ancestor of clade V. By further examining the orthologous genes shared by the autosomal ND element of *C. elegans* (clade V species) and clade III and clade IV species, all with the X + D translocation, we estimated that only ~60% of the *S. ratti* (clade IV) ND-derived genes' orthologs are also X-linked in *O. volvulus* or *B. malayi* (clade III), much less than the proportion of at least 85% (and up to 98%) of the NX-derived genes found among X-linked genes in pairwise comparisons among all of the Rhabditida species studied (Fig. 2b). This supports independent X + D translocations in the ancestors of the clade III or clade IV species. Similarly, the NB element corresponds to part of the X chromosomes of both *T. muris* (clade I) and *S. ratti* (clade IV), but fewer than half of the NB-derived orthologs are shared between these two species (Fig. 2b), also suggesting its independent translocation.

The Nigon element compositions of the X chromosomes of nematode species, in addition to the seven species studied above, suggest even more translocations (Fig. 2d, Supplementary Figs. 8–10, Supplementary Data 5). As only scaffold-level draft genome assemblies are available for these species, we estimated the coverage of their orthologous genes on different Nigon elements in male or pooled-sex genomic sequences, to identify patterns that indicate sex-linked regions (see above).

Among clade I species, in addition to *T. muris*, the inferred X-linked genes of two other *Trichuris* species (*T. trichiura* and *T. suis*) also include orthologs in *C. elegans* on both the NA and NB elements. X-linked genes of *Romanomermis culicivorax*, an outgroup to the *Trichuris* species, had orthologs only on the NA element (Fig. 2d, Supplementary Figs. 8b, 10). Most likely, therefore, the NA element was a sex chromosome pair in the ancestor of clade I nematodes and a translocation created the A + B state in the ancestor of *Trichuris*. If the NX element was the sex chromosome in the ancestor of all nematodes, a turnover event must have occurred, and resulted in the NA element becoming a sex chromosome in clade I.

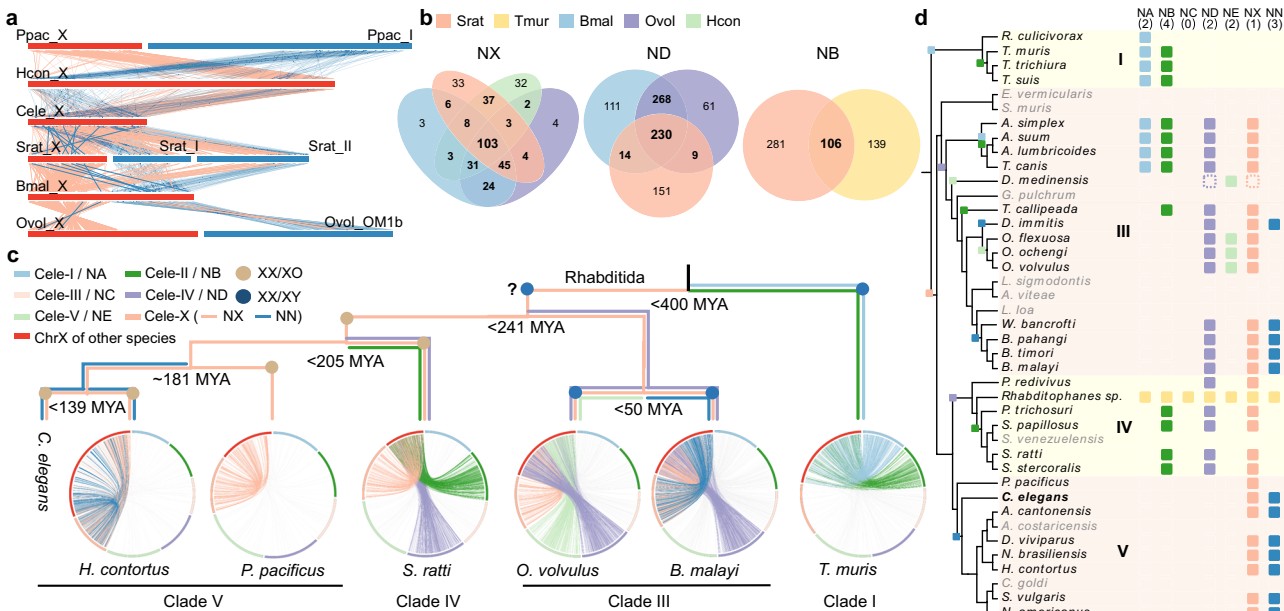

**Fig. 2 Evolution of sex chromosomes of nematode. a** Whole-genome alignments of sex chromosome sequences between six nematode species with chromosome-level genome sequence assemblies. The X chromosome of each species is indicated by a red bar, and blue bars represent the autosome numbers shown after the species names. The abbreviations are as follows: Srat: *S. ratti*, Tmur: *T. muris*, Bmal: *B. malayi*, Ovol: *O. volvulus*, Hcon: *H. contortus*. Each line represents an orthologous gene pair, with red lines indicating NX element genes and blue lines NN ones. **b** Venn diagrams[118] showing the numbers of shared sex-linked genes belonging to the same Nigon element in different species, with the same abbreviations as in (**a**). **c** Summary of the sex chromosome constitutions and turnovers among six nematode species with chromosome-level genome assemblies, other than *C. elegans*, organised according to the phylogeny. Nodes inferred to have had XY systems (which we infer to be the ancestral state, see the section of 'Evolution of sex-linked gene transcription' in the main text) are indicated with dark blue dots, and nodes with XO systems with light brown dots. The estimated divergence times[54] are also shown at the nodes. The X chromosomes of different species are indicated in red at the edge of each circos plot[119], and different colours indicate each Nigon element that contributes part of each species' X (these are also indicated by the distinct colours of the branches of the phylogeny). **d** Homologous Nigon elements detected in the sex chromosomes of all studied nematode species (except those shown in gray, where the contributions could not be determined); contributions are indicated by squares with the same colours as in part **c**. These results allowed us to infer the nodes at which different Nigon elements became part of the sex chromosome. The parentheses under the element names at the top indicate the number of times each Nigon element became part of a sex chromosome. *Rhabditophanes sp*.KR3021 (yellow squares) has no sex chromosomes. The dotted blocks represent reversion of sex chromosomes back to autosomes in *Dracunculus medinensis*.

Among the clade III species, we inferred that, after the ancestral X + D translocation in a clade III ancestor, the NA and NB elements were translocated to the X chromosome in the ancestor of the infraorder Ascaridomorpha (creating an X + D + A + B chromosome that is found in species including *Anisakis simplex* and *Ascaris suum*). An independent translocation of the NE element (X + D + E) also occurred in the ancestor of *Onchocerca* species including *O. volvulus*, and a translocation of the NN element (X + D + N) in the ancestor of both *B. malayi* and *Wuchereria bancrofti*. Other lineage-specific changes include translocations in *Thelazia callipaeda* (X + D + B) and *Dirofilaria immitis* (X + D + N). Interestingly, the X + D ancestral sex chromosome configuration has been replaced by the NE element in *Dracunculus medinensis* in a turnover event which occurred long enough ago to be detected by our coverage analysis (Fig. 2d, Supplementary Figs. 8a–10).

Among the clade IV species, we note that *Strongyloides* species are parasites, and have environmental sex determination in which XX females reproduce parthenogenetically in the parasitic stage, but can produce XO males in response to the host immune response. The mechanism involves loss of one copy of the X-linked sequences during oocyte mitosis[46,55], suggesting that this is a derived state in these species, and that the X is ancestral. Our inference of sex chromosomes in clade IV (Fig. 2d), including these species, suggested that, following the X + D translocation in the ancestor of clade IV species (Fig. 2c), a second translocation involving the NB element created the

(X + B + D) sex chromosomes in both the free-living *Parastrongyloides trichosuri* and all *Strongyloides* species studied here, as the sex chromosome of their outgroup species *P. redivivus* does not include NB element genes. It follows that these translocations occurred in a common ancestor of *P. trichosuri* and *Strongyloides*, consistent with the transition to parasitism, and environmental sex determination, in the latter being more recent.

We identified translocations only when the Nigon elements involved exhibited reduced male read coverage. Our analyses therefore inferred only regions that have been non-recombining for such a long evolutionary time that either Y-linked genes have been lost or have undergone so much sequence evolution that they do not map to their X-linked alleles. Importantly, however, the results suggest that (barring the unlikely possibility of insertion of large autosomal regions into a non-recombining part of the X) each of the translocation events led to a formerly autosomal region becoming completely sex-linked.

Despite the large number of rearrangements involving sex chromosomes that we identified, it is therefore currently not possible to ask whether sex chromosomes are more often involved in translocations than autosomes, because coverage analysis does not detect events involving only autosomes.

## Tests for evolutionary strata in nematodes with XY chromosome pairs. The time since a translocated Nigon element became non-recombining can be inferred if it still carries enough Y-linked

sequences to estimate the divergence level from their X-linked counterparts. If recombination becomes completely suppressed in a formerly recombining region, the region will start to diverge in sequence (as explained above)[56]. Events causing loss of recombination at different times are detectable from different levels of divergence (termed 'evolutionary strata') between sequences on the two sex chromosomes. Strata may either occur through successive chromosome inversions (e.g., some strata in mammals[24], birds[23] and sticklebacks[21]), or by other mechanisms affecting local crossover rates[56]. Strata could also arise on neo-sex chromosomes either (i) over time after their translocation (as in mammals), or (ii) immediately, as a direct consequence of the fusion in a species without crossing over in the heterogametic sex (as in *Drosophila*, see above). Unlike *Drosophila*, nematode males undergo crossovers. In both sexes, *C. elegans* chromosomes exhibit distinct recombination domains: a low recombination central region, flanked by high recombination 'arms', and very low recombination at the tips and other nematodes show similar patterns[57,58] (though exceptions are reported, including in *H. contortus*[59]). In *P. pacificus*, strong localisation of crossovers to one chromosomeend specifically in males was observed for elements NC and ND, though not for the three other autosomes[60]. If such a sex difference exists, it could prevent recombination across most of the genome in males, similar to the proposed situation in the guppy[61]. In species with a recombination landscape similar to that of *C. elegans*, an X-autosome fusion creates males heterozygous for the fused chromosome and the free autosome, and the protein machinery that ensures crossover localisation may reposition crossovers away from the fused chromosome end of the former autosome. Such a change was reported for an X-IV fusion in *C. elegans*[62], which has a decreased recombination rate in the fusion-proximal former chromosome arm region, but an increased rate in the distal region. To discover whether, and how, the fusions affected recombination on the neo-X chromosomes, and whether the cessation of recombination occurred in a single or multiple steps, we examined the translocated Nigon elements in different nematode lineages.

In extant clade IV and clade V species with XX/XO sex systems (Fig. 2c), the homologous chromosome (the 'neo-Y' chromosome) is no longer present, and only the translocated X-linked Nigon element (the 'neo-X') remains. In such systems, the only approach available to detect multiple recombination suppression events is to test for accumulation of repetitive sequences in the neo-X. Regions in which recombination was suppressed longest ago are likely to have higher accumulated repetitive content. However, deletions can occur in degenerated regions, and accumulation differences may be obscured by intrachromosomal rearrangements, so repeat distribution is unlikely to reliably reflect the pattern of distinct evolutionary strata. Not surprisingly, no clear evidence of evolutionary strata was found (e.g., in *H. contortus*, Fig. 3, Supplementary Fig. 4). The absence of neo-Y-linked genes shows only that these elements have completely stopped recombining, followed by genetic degeneration involving complete loss of genes, consistent with these neo-sex chromosomes having evolved long ago (Fig. 2d).

Three other clade I and clade III species have XX/XY sex systems with retained neo-Y chromosome sequences, and we expect to find neighboring regions belonging to different strata, reflecting their different Nigon elements, and different times during which these have been sex-linked. First, recently translocated Nigon element(s) are expected to be either pseudoautosomal regions (PARs), where recombination still occurs, or young strata. PAR boundaries with the adjacent fully sex-linked regions should be revealed by sharply increased male read coverage to levels similar to that of autosomal sequences. Second, strata that evolved at different time points, via

translocations or inversions, will have different Y-X sequence divergence levels (or, with less clarity, sex differences in SNP density), or, even less clearly, in the lengths of contiguously assembled Y-linked fragments. Repeat densities may also show increases at strata boundaries within fully sex-linked regions, particularly in the Y-linked haplotype (but also potentially in the X-linked one), as repeat elements are expected to accumulate rapidly after recombination stops[63]. We defined statistically significant transitions, and inferred strata boundaries, using change-point analyses (see 'Methods', Supplementary Fig. 11).

Across almost all X chromosomes of the two clade III species, *B. malayi* and *O. volvulus*, read coverage in females is nearly uniformly twice that in males, except at the end of the neo-X chromosome, where the lack of any coverage difference identifies a single probable PAR (or a very young stratum) in both species (6.3% of the total assembly length in *B. malayi*, and 12.9% in *O. volvulus*). Nearly half of the X chromosome length consists of intermingled ND- and NX-element derived genes (Fig. 3), indicating massive intrachromosomal rearrangements after the translocation of ND element genes to the NX element in the clade III ancestor (Fig. 2d). Similarly, the X chromosomes of *C. elegans* and *S. ratti* are also derived from two Nigon elements, and genes from the two elements are also intermingled (Fig. 2). The translocation boundary between the ND and NX elements in *B. malayi* or *O. volvulus* cannot be determined precisely due to the rearrangements, but we inferred in the previous section that the NX element must have been a sex chromosome for longer than the ND element. Therefore, each element should represent at least one stratum (labeled together as S0 + S1 in Fig. 3). S0 and S1 are shared by all clade III species, and S1 probably stopped recombining in males before the species divergence (estimated to be less than 50 MYA[64]; Fig. 2d).

The other X-linked genes are almost entirely derived from the NN element in *B. malayi*, and from the NE element in *O. volvulus*, neither of which is intermingled with ND or NX element sequences, consistent with more recent translocations compared with the ND element that formed the S1 stratum (Fig. 3, Supplementary Fig. 12). Parts of the ND, NE or NN elements may have continued recombining with their autosomal homologs after the translocation, as discussed above. However, their lower coverage in males than autosomes indicates that the translocated regions have now completely lost recombination and become degenerated Y-linked regions.

Nematode chromosome end regions often have high repeat content, probably indicating that the terminal regions recombine very rarely[57,58,60]. Interestingly, in both *B. malayi* and *O. volvulus*, the translocation breakpoint between the S0 + S1 region (ND + NX element) and the NE or NN element has a higher repeat content than other X-linked regions ($P < 2.2 \times 10^{-16}$, Chi-squared test), as do both ends of the fused chromosome (Fig. 3). It is not clear whether this reflects accumulation of repeats after recombination was reduced by a translocation in a formerly recombining chromosome end region, similar to the case of X + IV fusion in *C. elegans* mentioned above[62], or is simply a feature retained from the pre-translocation ancestral chromosome. If less distantly related species pairs differing by fusions can be found, it might be possible to distinguish between these possibilities.

Although few X-Y gametolog pairs were retained in either of *B. malayi* or *O. volvulus* (Supplementary Data 6), and Y-X divergence cannot be reliably estimated, there are some signs that recombination might not have been suppressed directly by the fusions (as explained above, recombination suppression is not expected, in contrast to the effect of X-A fusions in *Drosophila*). Evolutionary strata may therefore have evolved since the fusions occurred. As expected if strata younger than S0 + S1 subsequently

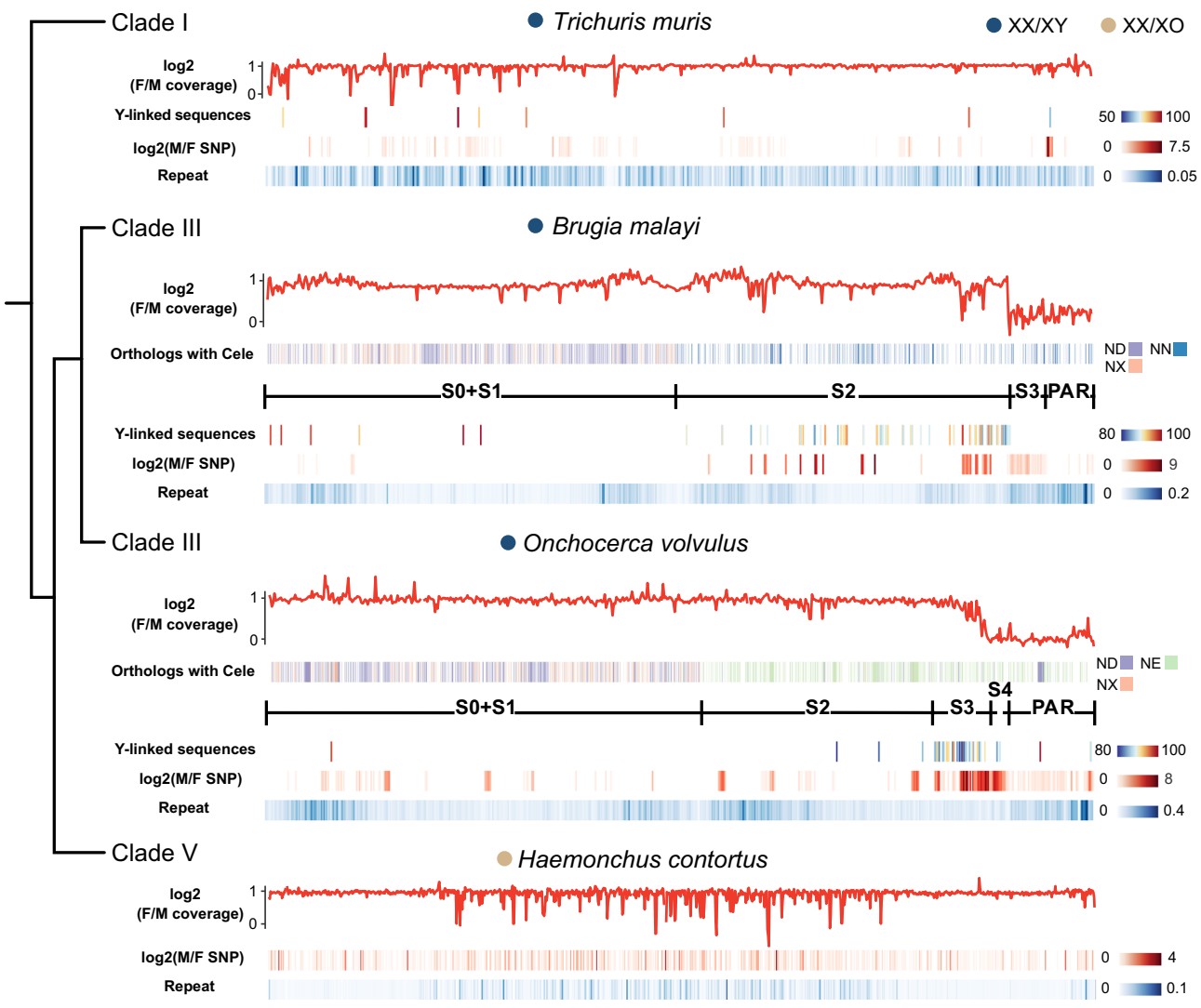

**Fig. 3 Evolutionary strata of four nematode sex chromosomes.** For the X of each species, the component Nigon elements correspond to strata inferred based on the presence of mappable Y-linked sequences, male vs. female ratios of mapped read coverage and SNP densities, and repeat densities. All metrics were estimated for 50-kb long, 10-kb overlapping windows. For *Brugia malayi* and *Onchocerca volvulus*, the orthologous genes (vertical lines) are colour-coded according to their different Nigon elements. For *Trichuris muris*, *B. malayi* and *O. volvulus*, the Y-linked sequences that could be assembled along the X are colour-coded according to their pairwise sequence divergence between the X/Y, and the log2 male vs. female SNP density ratios and the repeat density on the X are indicated by colour-codes. Strata are labeled S0 for the oldest stratum, followed by S1, S2, etc.). PAR stands for the pseudoautosomal region.

evolved, the NE or NN elements have more assembled Y-linked sequences (indicating less degeneration) than the ND + NX element region ($P < 2.2 \times 10^{-16}$, Chi-squared test). Moreover, the NN element in *B. malayi* can be divided into a stratum S2 region next to S0 + S1, based on genomic coverage indicating male hemizygosity (and pronounced gene loss from the Y), and a region with similar coverage in both sexes at the end of the chromosome (Fig. 3). The latter can be further divided into an S3 region, with a significantly higher SNP density in males than females ($P < 0.001$, Wilcoxon test, Supplementary Fig. 11), suggesting some divergence between the X- and Y-linked sequences without pronounced Y gene loss (Fig. 3), and a PAR with no SNP density difference between the sexes. Overall, we conclude that the *B. malayi* sex chromosomes could have at least four evolutionary strata and a PAR. A similar approach suggests five evolutionary strata and a PAR in *O. volvulus* (Fig. 3, Supplementary Fig. 11). Only 1% of *O. volvulus* X-linked S2 (element NE) sequences have corresponding Y-linked sequences,

whereas the S3 carries 12% of the genes found in the counterpart X-linked region, suggesting that it formed more recently.

Analysis of the lengths of contiguously assembled Y-linked fragments failed to identify either PARs or evolutionary strata in the XY sex systems of the clade I species *T. muris* and *T. suis*. Only 0.06% of the *T. muris* X-linked sequences have Y-linked counterparts. Their Y chromosomes have probably become completely degenerated, as suggested by a previous cytogenetic study of *T. muris*[65], so that few fragments can be assembled.

**Evolution of sex-linked gene transcription.** Given that non-recombining regions have recently formed between sex chromosomes of both nematodes and schistosomes, it is interesting to ask whether genes in these regions followed the evolutionary trajectory of canonical ancestral sex chromosomes, including evolution of dosage compensation (DC), and a different distribution of sex-biased genes relative to autosomes[66,67].

Genetic degeneration of Y-linked genes is expected to reduce transcript abundances of hemizygous X-linked genes below the ancestral level, favouring the evolution of dosage compensation (DC), either in individual genes (incomplete DC, iDC) or of the entire X chromosome (chDC)[68]. Dosage compensation likely initially involves an upregulation of X-linked genes in males (as in Drosophila[69]), but this may be followed by downregulation in females (as in mammals[70] and C. elegans[71]). The opposite changes are expected in species with ZW systems[68]. S. mansoni was reported to have chDC in the cercarial stage, but iDC after sexual differentiation[20,72]. Meiotic sex chromosome inactivation (MSCI) in male gonads can also lead to evolution of an underrepresentation of male-biased genes on X chromosomes, as reported in C. elegans[73] and Drosophila[74]. Because of its inheritance pattern and selection differences between the sexes[66,75], the X chromosome is also expected to accumulate genes with female-biased expression ('feminisation') and become depleted for genes with male-biased expression ('demasculinisation') if the mutations are expressed in heterozygotes (wholly or partially dominant), while the Y chromosome should become 'masculinised'. These changes occur either through expression changes, or relocation of individual genes between chromosomes. Both feminisation and demasculinisation have been reported for numerous insects[29,74] and some nematode species[76], and affect relative expression levels of sex-linked and autosomal genes, in addition to the effects of MSCI and dosage compensation.

Comparisons of the transcriptomes from whole adults between fully X-linked versus autosomal genes in males (X/A ratios), and between the sexes (M/F ratios for X-linked genes), suggest very diverse forms and extents of DC in our seven representative nematode species (Fig. 4a, b). All the nematode species studied showed X/A expression ratios below 1 in males (based on between 782 to 2742 X-linked genes and 5398 to 10,490 autosomal genes, depending on the species, and Wilcoxon tests yielding $P < 0.05$ for all species, see Supplementary Data 7). These patterns were the same, regardless of whether we included strongly sex-biased genes ($\geq$2-fold transcriptional difference between sexes) or not (Supplementary Fig. 13). This supports the genomic coverage data showing degeneration of the Y chromosome, though, because the tissues analysed included gonads, MSCI of the male X chromosome may also contribute (Fig. 4a).

However, in clade I and clade III species, most X-linked genes in females, despite having twice the genomic copy number compared with males, exhibited whole-body transcription levels similar to, or even lower than, in males. This suggested either upregulation of X-linked gene expression in males, or a down-regulation in females for nematodes in these clades (Fig. 4a). While autosomal genes have the same genomic copy numbers between sexes, they are expected to exhibit an equal transcription level between sexes. However, autosomal genes of all studied species of these two clades exhibited a significantly ($P < 0.05$, Wilcoxon test) lower transcription level in females than in males. Particularly in O. volvulus, the median transcription level of autosomal genes was four times higher in males than in females. In addition, the transcription levels of autosomal genes in male O. volvulus were similar to those of their orthologs in B. malayi or other species in clade I groups (see 'Methods', Fig. 4c, Supplementary Fig. 14). These findings suggested that genome-wide downregulation of transcription levels in females not restricted to the X chromosome, probably accounted for a similar or even male-biased transcription pattern on the X chromosome (Fig. 4a, Supplementary Fig. 14). The female-biased sex ratio of O. volvulus[77] may reflect selection for lower female transcription to resolve 'sexual conflict', as suggested in ref. [75]. Intriguingly, in both B. malayi and O. volvulus females, estimated M/F transcription ratios of X-linked S0 region genes were closer to 1 than for autosomal genes (Fig. 4c) and appeared to show iDC.

In the clade IV and clade V species studied, however, transcription levels of X-linked genes in females were lower than those of autosomal ones (Fig. 4b), except for H. contortus, suggesting a chDC mechanism similar to that in C. elegans. All clade IV and clade V species' M/F transcription ratios were, nevertheless, lower for X-linked genes than autosome ones probably due to MSCI in males[66,75].

The X chromosomes of most nematode species studied consistently showed a depletion of male-biased genes (using the criterion of 2-fold higher transcriptional levels in males than in females), and an enrichment in female-biased genes, compared to autosomes (all species $P < 0.05$ by Chi-square tests; Fig. 4d). Feminisation of the O. volvulus X chromosome seems to have evolved more slowly than demasculinisation, as all evolutionary strata have become demasculinised, but only S0 is enriched in female-biased genes. The Y chromosomes of both O. volvulus and B. malayi show signatures of masculinisation, involving recruitment of gene duplications from autosomes. Among the total 91 O. volvulus and 219 B. malayi Y-linked genes (Supplementary Data 4), we identified 10 and 13 Y-linked genes, respectively, with no homologs elsewhere in the genome, but with an autosomal homolog in the other species as their potential progenitors, and these mostly have male-biased transcript levels (Fig. 4e). This suggests that the Y chromosomes of both species have preferentially fixed relocated autosomal genes with pre-existing male-related functions (Supplementary Fig. 15). Among the other Y-linked genes, we found two genes (Bm2848: WBGene00223109 and Ovo-sma-5: WBGene00242293) with X-linked homologs in the S0 region of the NX element. This suggests that the ancestor of clade III species had a Y chromosome homologous to the NX element (Supplementary Data 8), which has now become degenerated. The sex chromosome system of the nematode or ancestor of the Rhabditida ancestor might therefore have been XX/XY rather than XX/XO (Fig. 2c), contradicting the currently accepted view[42,43].

Finally, we also found evidence of masculinisation of the S. mansoni Z chromosome. The overall median expression level of S. mansoni Z-linked genes in testes was significantly higher ($P < 0.05$, Wilcoxon test) than that of autosomal genes (Supplementary Fig. 16).

**Transitions between sex systems in flatworms and nematodes.** Having defined the ancestral sex-determining regions in which suppressed recombination first arose in schistosomes (Supplementary Fig. 1), we used the available somatic and gonad tissue transcriptome data of the (hermaphroditic) trematode Clonorchis sinensis (liver fluke)[78], a relatively close outgroup to schistosomes, and the (hermaphroditic) taeniid cestode Taenia multiceps[79], to suggest transcription patterns in the hermaphroditic ancestor of flatworms before gonochorism evolved. C. sinensis and T. multiceps were estimated to have diverged from the gonochoristic schistosomes about 70 and 100 MYA, respectively[53,79]. We first examined the chromosomal rearrangements that produced the extant schistosome sex chromosomes, then looked for candidate sex-determining genes that might be responsible for the transition to gonochorism, and finally studied genome-wide transcriptional changes after the transition. Our inferences of genes potentially participating in the sex-determination pathways of flatworms and nematodes other than C. elegans used the well-characterised C. elegans genes as a reference. However, it is known that sex-determining pathway genes can undergo rapid turnovers between even related species[80,81]; therefore, orthologs of C. elegans genes in other nematode species, and particularly the deeply diverged flatworms do not necessarily have a sex-determining function,

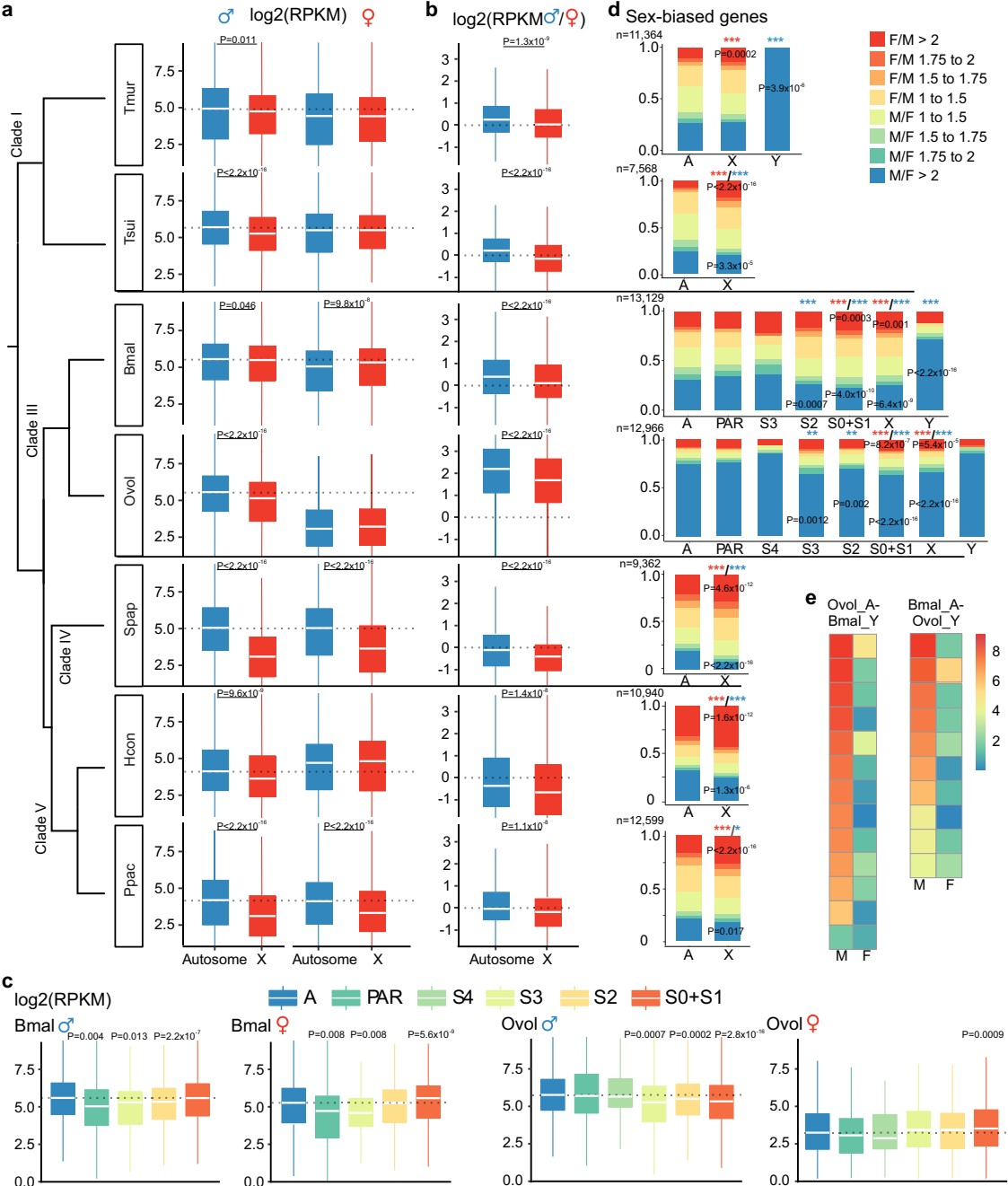

**Fig. 4 Evolution of sex-linked gene transcription in nematodes. a** Transcription levels of autosomal (blue) and X-linked genes (red) of different nematode species in males and females. The dotted line shows the median log2 normalised transcription levels of autosomal genes. **b** Male/female transcription ratios of autosomal (blue) and X-linked genes (red) in nematodes. **c** Transcription levels of *B. malayi* and *O. volvulus* genes divided into different chromosome regions, with autosomes (A) in blue, pseudoautosomal regions (PAR) in green, and different evolutionary strata (S0-S4) in different colours. The dotted line is the median log2 normalised transcription level of autosome genes. Overall the whole-genome transcription level is male-biased. The two-sided Wilcoxon rank-sum tests yielding *P* < 0.05 in all species (based on between 782 to 2742 X-linked genes and 5398 to 10,490 autosomal genes, see Supplementary Data 7). The boxplots show the 25th percentile, median, and 75th percentile, and whiskers are set within 1.5 times the interquartile range. **d** Percentages of male- (blue) and female- (red) biased genes, showing, respectively, deficiency and enrichment on the X relative to the autosomes across the studied nematode species (based on between 7568 to 13,129 genes). Significant deficiencies or enrichments by two-sided Chi-square tests are indicated with blue asterisks (male-biased gene), and with red asterisks (female-biased gene): *P < 0.05; **P < 0.01; ***P < 0.001. **e** Transcription patterns for autosomal *O. volvulus* genes (Ovol_A) that are homologous to the Y-linked genes of *B. malayi* (Bmal_Y) in males (M) and females (F), or vice versa. Species abbreviation: Tmur: *T. muris*, Tsui: *T. suis*, Spap: *S. papillosus*, Bmal: *B. malayi*, Ovol: *O. volvulus*, Hcon: *H. contortus*, Ppac: *P. pacificus*.

and need to be functionally validated in the focal species in the future. In the following, we mainly focus our description on certain genes that have functional evidence in both schistosomes and *C. elegans*.

Alignments of sequences of the chromosomal assemblies of *S. mansoni* with *C. sinensis* (Fig. 5a, Supplementary Fig. 17a) and *T. multiceps* (Supplementary Fig. 17b) indicate that the *S. mansoni* Z chromosome evolved through a translocation, as first proposed in

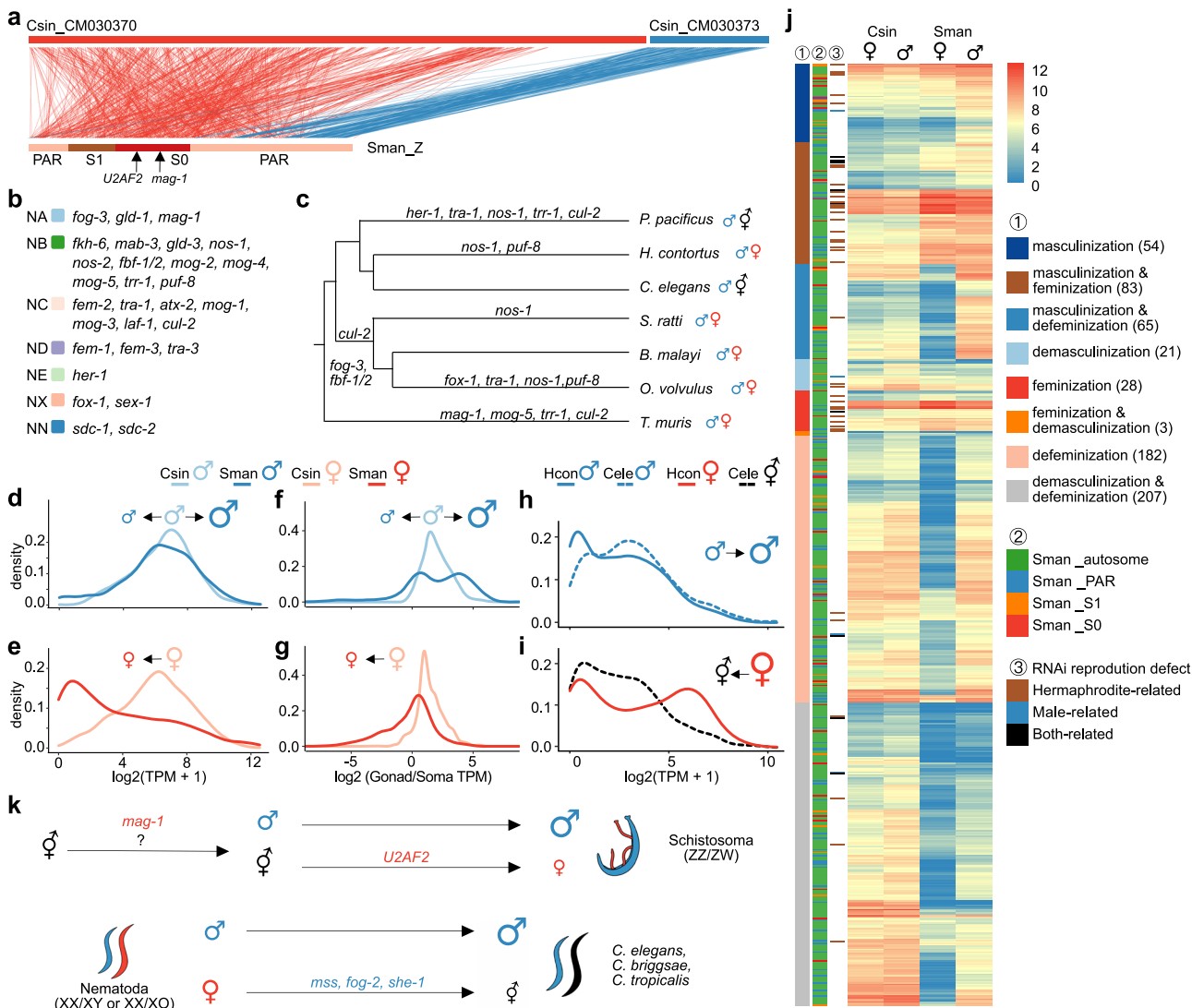

**Fig. 5 Transitions of sexual systems in platyhelminths and nematodes. a** Gene synteny relationships between *Schistosoma mansoni* chrZ (Sman-chrZ) and *Clonorchis sinensis* linkage group CM030370 (Csin_CM030373, red) and CM030373 (Csin_CM030373, blue). The evolutionary strata of *S. mansoni* are shown in different colours, and the positions of the candidate sex-determining genes *U2AF2* and *mag-1* on the chrZ are indicated by arrows. **b** The distribution of the reported genes in the *C. elegans* sex determination pathway on different Nigon elements. **c** Lineage-specific duplication of *C. elegans* sex-determining genes across the phylogeny of nematodes. **d** Density plots comparing the testis or ovary (**e**) transcription levels, or the testis (**f**) or ovary (**g**) transcription specificity (measured by the transcript level ratios of gonad vs. soma) of gonad-biased genes of *C. sinensis* (light blue or red) and their *S. mansoni* orthologs (dark blue or red). Masculinisation, i.e., increase of testis transcription levels, is indicated with a larger male sign, and vice versa for feminisation and demasculinsation etc. Comparison of whole-body male (**h**) and female (or hermaphrodite, **i**) transcriptomes between *C. elegans* (dotted line) vs. *H. contortus* (solid line). **j** Masculinised *S. mansoni* ortholog of *C. sinensis* gonad-enriched genes (defined as the former having at least 2-fold higher transcription level in testis than the latter), and demasculinised, feminised, or defeminised genes. Their testis or ovary transcription levels of each gene are shown in the heatmap, as well as its functional category, chromosome location in *S. mansoni*, and information about the knockdown phenotype of its *C. elegans* ortholog (Supplementary Data 10). **k** A proposed model for the transition of sexual systems in platyhelminths and nematodes. We hypothesised a 2-step model for the transition from a hermaphroditic ancestor to the gonochoristic schistosomes (see text). The recent independent transitions from gonochorism to androdioecy in different *Caenorhabditis* species have been studied previously[88],[120], as have the genes reported here. Changes in gonad transcription levels in relation to the sexual systems are indicated using larger or smaller male or female/hermaphrodite symbols.

the 1980s[41]. Our analyses clarified that this involved two ancestral chromosomes homologous to two (CM030370 and CM030373) of the seven *C. sinensis* linkage groups, and to LG3 and one part of LG1 of *T. multiceps*, followed by intrachromosomal rearrangements (Supplementary Fig. 17b). These rearrangements might have led to the suppressed recombination of the oldest schistosome stratum, S0, which is expected to carry the candidate sex-determining genes of all *Schistosoma* species, since the translocation boundary overlaps the *S. mansoni* S0/PAR

boundary. Within the Z-linked S0 region of *S. mansoni*, we found the ortholog of the *C. elegans* gene *mag-1*, whose knockdown in *C. elegans* causes the hermaphrodites to produce sperm instead of oocytes[82]; in *S. japonicum*, its knockdown causes apparent cell proliferation in testicular lobes[83]. Functional experiments in *Drosophila*[84] have indicated an important and evolutionarily conserved role of *mag-1* during oogenesis. If gonochorism originated in a non-gonochoristic schistosome ancestor via two mutations, a possible scenario might be that

recessive female-suppressing mutations have first affected *mag-1's* likely feminising function in the germline, leading to a transition from hermaphroditism to androdioecy as the first step in the evolution of gonochorism[2]. The second mutation might have given rise to the dominant demasculinising/feminising gene *U2AF2*, which was recently identified on the S0 region of W chromosome of *S. mansoni*[85,86].

The schistosome Z-linked S0 region also had a duplication of *fox-1*. In *C. elegans*, *fox-1* 'communicates' the different ratios of X vs. autosome copy numbers between sexes to the downstream switch gene *xol-1*, and directs male or hermaphrodite development[87]. However, the duplication of *fox-1* in the S0 region of the Z chromosome is specific to *S. mansoni* and has not been detected in *S. japonicum* or *S. haematobium*. Therefore, the *fox-1* paralog of *S. mansoni* may not have a sex-determining function.

Duplication of orthologs of genes involved in the sex-determination pathway of *C. elegans*[88] also seems to have occurred in other nematodes, as we found lineage-specific duplications of 12 sex-determination pathway genes on different Nigon elements (Fig. 5c). For instance, the ortholog of *xol-1*, the direct target gene of *fox-1*, is present only in *Caenorhabditis* species[89,90] but not in the other nematode species (or in the schistosome species studied here, Supplementary Data 9). The *fog-3* gene is critical for spermatogenesis in *C. elegans*[91], and is duplicated in both *B. malayi* and *O. volvulus* (Supplementary Fig. 18), probably in the ancestor of clade III nematode species. Interestingly, *O. volvulus* also has a duplication of *fox-1*, similar to *S. mansoni*.

We found no clear relationship between the numbers of sex-determining gene orthologs and the probability of particular Nigon elements being translocated to an ancestral sex chromosome pair. The NB element carries the largest number of orthologs of *C. elegans* sex-determining genes and has been more frequently translocated on to the ancestral sex chromosome than other Nigon elements (Fig. 2d). However, the NC element, with orthologs of seven *C. elegans* sex-determining genes, was not involved in such a translocation in any nematode species so far studied.

Next, we compared the transcriptomes among the gonochoristic *S. mansoni* vs. the hermaphrodites *C. sinensis* and *T. multiceps*, and between the gonochoristic *H. contortus* vs. androdioecious *C. elegans* (Fig. 5d–i, Supplementary Fig. 19), in order to reveal genome-wide changes of transcription following transitions in sex systems. We focused our comparison on gonad tissues whose transcriptomes were available for the three flatworm species and are probably more likely to change than somatic tissues in response to the transition. We defined gonad genes as those with at least 2-fold higher transcription levels in gonads than the soma. For the two nematode species, we analysed sexed whole-body transcriptome data, as tissue-specific transcriptomes were not available.

*C. sinensis* and *T. multiceps* gonad genes had unimodal distributions of transcription levels and specificities (measured by the gonad/soma transcription ratios); in contrast, their *S. mansoni* orthologs exhibited bimodal distributions of both values in testes, predominantly with lower values of both measures in ovaries (Fig. 5d–g, j, Supplementary Figs. 19, 20). In particular, there were 202 *S. mansoni* genes that became masculinised (i.e., had increased testis transcription levels) relative to *C. sinensis*, compared with 231 genes that became demasculinised (Fig. 5j). In contrast, only 114 genes became feminised (i.e., had increased ovary transcription levels) compared with 389 defeminised genes (Fig. 5j). A similar pattern is observed in comparisons to the more distantly related *T. multiceps* vs. *S. mansoni* (Supplementary Figs. 19, 20). These transcriptome patterns suggest that the

transition from a hermaphroditic ancestor (represented by *C. sinensis* or *T. multiceps*) to the ZW system in the gonochoristic *S. mansoni* involved mostly defeminisation, a lesser or similar extent of masculinisation and demasculinisation, and an even lower extent of feminisation of gonad gene expression. *C. elegans* orthologs of feminised genes in *S. mansoni* are enriched for mutant phenotypes related to female reproduction (e.g., 'egg laying', 'vulva development') and embryonic or germ cell development (e.g., 'embryonic lethal' and 'germ cell development variant') ($P < 0.05$, Chi-square test, Supplementary Data 11). While male-related mutant phenotypes (e.g., 'male behaviour variant' and 'male response to hermaphrodite variant') show enrichment only in masculinised, but not in feminised, genes (Supplementary Data 12). Interestingly, we identified 65 *S. mansoni* genes that were both masculinised and defeminised, and 3 genes that were feminised and demasculinised relative to *C. sinensis* in the gonads, suggesting possible resolution of 'sexual conflict' after the evolution of gonochorism (Fig. 5j).

Similarly, the reverse transition from a gonochoristic ancestor represented by *H. contortus* in relation to the androdioecious *C. elegans* seems mainly to have involved defeminisation and masculinisation of the gene transcription (Fig. 5h, i, Supplementary Figs. 21, 22).

## Discussion

The evolution of sex chromosomes in some taxa involves the primary transition from a hermaphroditic system or from environmental sex determination to a dioecious/gonochoristic species with genetic sex determination. Such transitions usually are accompanied by potential suppression of recombination in and around the sex-determining gene(s), and the recombination suppression sometimes even extends to other sex chromosome regions. In other species, turnover events may create new sex-determining regions, which may also evolve into non-recombining regions. In either case, autosomal regions that fused or translocated to sex chromosomes may sometimes also become completely sex-linked, either in species that lack recombination in the heterogametic sex[26], or potentially in species with recombination in both sexes through subsequent recombination loss.

Schistosomes have evolved gonochorism from hermaphroditism and exhibit strong morphological sexual dimorphism (see above). As explained earlier, the first step in such a primary transition must either involve a mutation creating females (producing a gynodioecious population), or one creating males (producing an androdioecious population). A mutation in the highly conserved oogenesis-related or feminising gene *mag-1* might have produced males in the ancestor of schistosomes. However, it seems unlikely that such a mutation could have greatly increased male fitness, compared with that of the ancestral hermaphrodite, as required for the establishment of androdioecy. Females could have arisen due to a dominant mutation in the reported W-linked candidate sex-determining gene *U2AF2* (Fig. 5k)[85,86]. The involvement of these genes in the evolution of schistosome sex-determination needs to be tested in the future.

Our finding that the evolution of the present schistosome sex-linked regions was followed by transcriptional changes of many genes in gonads (Fig. 5d–g) is consistent with the hypothesis of sexual antagonism in the hermaphroditic ancestor, favouring re-allocation of resources after separate sexes evolved[75]. Assuming that higher transcription levels reflect advantageous changes, the results in schistosomes suggest that conflicts were resolved and a new optimum reached more frequently in males (masculinisation and overwhelming defeminisation) than in females (feminisation) in the gonads. In the transition to dioecy in *Silene latifolia*, with

male, instead of female heterogamety, and an XY sex chromosome system, transcriptional changes occurred most frequently in females[92]. The results suggest transcriptional changes after the X or Z chromosome became hemizygous in one sex, resulting in masculinisation of the schistosome Z (as shown in Supplementary Fig. 16) and feminisation of the X chromosome in *S. latifolia*.

Following the origin of gonochorism, both schistosome and nematode ancestral sex chromosomes have undergone translocations of autosomes, like those in many other taxa[26,56] (Figs. 3 and 5a). The translocated autosomes, or large parts of them, have become completely non-recombining in both phyla, and in nematodes they have become strongly degenerated like the ancestral sex chromosomes. How loss of recombination happened is an interesting question. Recombination between the autosomes involved in a fusion or translocation with sex chromosomes often maintain autosomes' former recombination patterns. However, the study of fusions between the *C. elegans* X chromosome and chromosome IV[62] suggested that crossovers may be re-positioned away from the fusion junction, creating a new chromosome with two arm regions (whereas the two participating chromosomes each contained two arm regions). In the fused chromosome, a potentially large former arm region close to the fusion point may thus have greatly reduced recombination, and if the fusion involves the X chromosome, this will occur specifically in males. Such events can therefore create new sex-linked regions without involving selection for suppressed recombination.

Translocations of autosomes to ancestral sex chromosomes may be common in nematodes because some, though not all, nematodes have holocentric chromosomes[93]. Such chromosomes may be more prone to fusions or fissions than monocentric ones, in which such rearrangements may lead to a loss or multiplication of centromeres[94]. However, a recent comparison of insects with different centromere types found no evidence supporting this hypothesis[95].

Finally, we annotated many Y- or W-linked genes additional to those already known in the flatworm and roundworm species studied here (Figs. 4 and 5). We also found homologs of *C. elegans* sex-determination pathway genes that may have undergone duplications in different nematode species. These genes could be involved in the divergence of the sex-determination pathways, as has already been documented between *C. elegans* vs. *C. briggsae*[96]. Functional verification in other nematode species is needed in the future. The present study could not identify further changes that may have occurred after lineage-specific duplications of these genes, and possible changes also need to be studied further. For example, *gld-1* was independently recruited into the sex-determination pathways of *C. elegans* and *C. briggsae*; in *C. elegans* it acts to promote spermatogenesis, but it promotes oogenesis in *C. briggsae*[97]. Its co-factor, *fog-2*, evolved by a duplication and acquisition of a new GLD-1-binding domain in *C. elegans*[88,98]. The newly annotated, candidate sex-determining genes could be a useful resource for future studies of parasite control through interfering with their sexual life cycles.

## Methods

**Identification and annotation of sex-linked regions**. The genomic and transcriptomic data sets used in this project were retrieved from WormBase ParaSite (https://parasite.wormbase.org/index.html) and NCBI (all accession numbers are included in Supplementary Data 1, 2), based on the published phylogeny of flatworms and nematodes[99]. We improved several contig- or scaffold-level genome assemblies with RaGOO[100], using chromosome-level genome assemblies of closely related species as references. We retained only RaGOO assemblies with assembled sizes of larger than half of the input genome size (Supplementary Data 3). We then aligned the male or female Illumina reads to the respective genomes using Bowtie2 v2.2.9[101] and quality-filtered the alignments with SAMtools v1.11[102] (-q 33), and kept only uniquely aligned reads.

To estimate male or female read coverage along chromosome or scaffold sequences, we divided the genomes into 10 kb non-overlapping windows with

BEDTools v2.27.1[103]. When a bimodal read coverage distribution of all windows was detected, we normalised the peak of the higher coverage level (autosomal sequences) to be centered at 1. We expected sequences whose coverage levels were concentrated at 0.5 to represent X- or Y-linked sequences in nematodes, or Z- or W-linked sequences of schistosomes. For species that did not have genomes assembled to the chromosomal level, we required at least 70% of the windows of a given scaffold or contig to be sex-linked for the entire scaffold or contig to be classified as sex-linked. For species with chromosomal-level genomes, we plotted the male or female read coverage values along the entire candidate sex-chromosome sequence for manual inspection (Supplementary Figs. 3, 4). Specifically, the W- or Y-linked sequences were separated from the Z- or X-linked sequences based on an expectation that the former can be mapped exclusively by female or male reads, respectively. For candidate Y-linked sequences, we required their male *vs.* female read coverage ratios to be >5, and the normalised coverage level to be >0.2 in males, and <0.2 in females. Similar cutoffs were applied to identify W-linked sequences in *S. mansoni* (Supplementary Fig. 5).

To identify Y-linked genes that are not in the genome assembly, we aligned male RNA-seq reads to the female genome assembly using Tophat v2.1.1[104]. We then de novo-assembled the unmapped male reads, which should be enriched for Y-linked genes, using Trinity v2.4.0[105]. For the resultant transcriptome assembly, we then discarded the transcripts which could be aligned (with >50% of the length aligned, and >50% alignment identity) to the female genome assembly using nucmer v3.23[106]. We also discarded sequences that could be aligned by female RNA-seq reads, with >50% of the transcript lengths aligning, using Bowtie2 v2.2.9[101]. Then, for the remaining male transcript sequences, we assessed their male- and female-read coverage using Illumina genomic reads. We identified a transcript as Y-linked, if its male vs. female genomic read coverage ratio was >5, with the normalised female genomic coverage of <0.5, and a male genomic coverage of >0.1. Then we predicted the coding regions within these Y-linked transcript assembly sequences using TransDecoder (https://github.com/TransDecoder/TransDecoder). We applied a similar method and cut-off to identify the additional *S. mansoni* W-linked genes[107]. The Y- and W-linked identified genes are summarised in Supplementary Data 4.

**Assignment of Nigon elements**. We downloaded the protein annotation files from WormBase (Supplementary Data 1) and used the longest protein to represent each gene. We identified the one-to-one orthologs of *C. elegans* genes in the chromosome-level assemblies of six representative nematode species (*T. muris, B. malayi, O. volvulus, S. ratti, H. contortus,* and *P. pacificus*) and *S. mansoni* using OrthoFinder v2.2.6[108]. *C. elegans* autosome chrI to chrV were previously assigned to the Nigon elements NA to NE, and the X chromosome was identified as carrying orthologs of the Nigon elements NX and NN[36]. We then annotated the respective Nigon element for each gene in the other nematode species according to the Nigon information of their ortholog in *C. elegans*. Specifically, we assigned NX element membership to the X-linked genes of *C. elegans* homologous to those of *P. pacificus*, and NN membership to the remaining *C. elegans* X-linked genes homologous to *P. pacificus* chrI genes. The annotated genes in all species studied, with their assigned Nigon element membership are shown in Supplementary Data 5.

To identify the Nigon elements that are sex-linked in individual species, we first estimated the read coverage of each gene in males (Supplementary Fig. 8). Genes with normalised male read coverage between 0.4 and 0.6, or a mixed sex read coverage between 0.4 and 0.8 were annotated as putatively sex-linked. The percentage of sex-linked genes in each clade III species was then estimated, and Nigon elements assigned for the sex-linked genes (Supplementary Fig. 9).

**Analyses of evolutionary strata**. We combined multiple types of information to distinguish between the PARs and the differentiated completely sex-linked regions, and to infer different evolutionary strata. These included Nigon element composition, SNP density and coverage of reads in males and females, sequence divergence between alignable X- and Y-linked sequence pairs that could be identified, and the length of assembled Y-linked sequence. We aligned the reads to the assembled chrX sequences using BWA v0.7.17[109], and calculated their coverage per 50 kb- using 10 kb-overlapping windows employing BEDTools v2.27.1[103]. We identified the SNPs using the Genome Analysis Toolkit (GATK 3.8)[110] and annotated repeats employing RepeatMasker v4.07[111]. Read coverage, and SNP and repeat abundances were then visualised in 50 kb windows of the complete X chromosome assembly of the species investigated.

We identified candidate PARs or extremely young strata as continuous X-linked regions with log2 coverage ratios between the sexes between −0.5 and 0.5. Change-point analysis (described below) were used to identify the boundary between the candidate PAR and sexually differentiated regions using the location where the coverage ratio changed statistically sharply (Fig. 3, Supplementary Figs. 4, 11). Some X-linked regions exhibited clearly higher SNP densities in males within the region where coverage was similar in both sexes, and such regions were classified as candidate young evolutionary strata (e.g., the S3 of *B. malayi* and S4 of *O. volvulus*). We defined the boundary between S0/S1 and S2 in *B. malayi* and *O. volvulus* by plotting the Nigon element information of all the genes along the chromosome. This revealed a pronounced separate distribution of NX + ND element genes vs. those assigned NN or NE membership. The boundary between S2 and S3 of *O. volvulus* was defined based on different lengths of assembled Y-linked sequences

and SNP densities between sexes (Fig. 3, Supplementary Figs. 4, 11). The *O. volvulus* S3/S4 boundary and the *B. malayi* S2/S3 boundaries were inferred based on sharp changes in male read coverage; the *O. volvulus* S4 and *B. malayi* S3 regions exhibited higher SNP densities in males, compared with the respective PAR regions ($P < 0.001$, Wilcoxon test, for both species). For *S. mansoni*, whose S0 was expected to be shared with *S. japonicum*, we defined S0 boundary with S1 using homologous genes in the two schistosome species (Supplementary Fig. 4)[41].

A change-point analysis detects the data point showing a significant shift of mean values between two consecutive datasets, which could be time-series data, or in this work, certain values of many consecutive genomic windows. All boundaries were determined by change-point analyses using the cpt.mean function in the R Changepoint (v.2.2.2)[112] package with the 'BinSeg' method and Akaike information criterion (AIC) penalty. The maximum number of changepoints was set to 12 ($Q = 12$), to uncover significant changes in the female vs. male coverage, and male vs. female SNP densities in 50-kb windows across the *B. malayi* and *O. volvulus* X chromosomes (Supplementary Fig. 11). To define the *O. volvulus* S3/S4 boundary, we chose the genomic window that exhibited the lowest female/male coverage ratio defined by the change-point analyses. The *B. malayi* S3/PAR and *O. volvulus* S4/PAR boundaries were defined similarly (Supplementary Fig. 11).

**Gene transcription analysis**. We mapped the RNA-seq reads to the reference genome using HISAT2 v2.0.4[113], counted the aligned read number employing HTSeq v0.11.0[114], and calculated the normalised transcription levels in RPKM (reads per kilobase of transcript per million mapped reads)[115] (Fig. 4, Supplementary Figs. 1, 4, 13, 14, 16 and 18) or TPM (transcripts per million) (Fig. 5, Supplementary Figs. 19–22). For cross-species comparisons, only 1-to-1 orthologs defined by OrthoFinder v2.2.6[108] that are located on autosomes or sex chromosomes in both species are considered between any two species, and their RPKM were compared (Fig. 4c, Supplementary Fig. 14). The RNAi phenotype data of genes (Fig. 5j, Supplementary Fig. 20) was obtained on website (https://wormbase.org/tools/mine/simplemine.cgi), and enriched phenotype (Supplementary Data 11, 12) was obtained from the modPhEA (http://evol.nhri.org.tw/phenome2/).

**Phylogenetic trees of selected genes**. We used the longest transcript to represent each gene for reconstructing ML (maximum likelihood) phylogenetic trees. These analyses were done to identify Y-linked genes that may have arisen by duplications onto the Y, including ones with potential autosomal progenitor homologs in other species (Supplementary Figs. 15, 18) employing MEGAX v10.1.1[116].

**Reporting summary**. Further information on research design is available in the Nature Research Reporting Summary linked to this article.

## Data availability
The genomic and transcriptomic data sets used in this project were retrieved from WormBase ParaSite (https://parasite.wormbase.org/index.html) and NCBI (https://www.ncbi.nlm.nih.gov/). A full list of accession IDs is available in the Supplementary Data 1, 2. Source data are provided with this paper.

## Code availability
All custom codes used in this work are available at https://github.com/Flyase?tab=repositories.

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

## Acknowledgements

We thank the three anonymous reviewers and Dr Neil D. Young for their helpful comments on the manuscript. Q.Z. is supported by the National Natural Science Foundation of China (32170415, 32061130208), the Natural Science Foundation of Zhejiang Province (LD19C190001) and the European Research Council Starting Grant (grant agreement 677696). R.B.G.'s research is partly supported by the Australian Research Council (ARC).

## Author contributions

Q.Z. conceived the project, Y.W. and D.C. performed the analyses, Q.Z., D.C., Y.W., and R.G. wrote the paper together.

## Competing interests

The authors declare no competing interests.
