## [Peer Review File · Nature Communications]

Evolution of sexual systems, sex chromosomes and sex-linked gene transcription in flatworms and roundwormsReviewers' Comments:

Reviewer #1:

Remarks to the Author:

The authors collected publicly available genome sequence data and characterized sex chromosomes of flatworms and roundworms. Although I find some data interesting, I cannot follow the methods or their reasoning in several parts.

It is not clear how they determined the sex chromosome systems (Fig. 1). According to the methods, they used the coverage data (some examples are shown in Fig. 2b and all data are shown in Fig. S1). I cannot understand how they can determine the sex chromosome systems based on these data. For example, *B. pahangi* is hemephrodite, but there are two peaks. *S. mattheei* was judged as ZW, but the data shown in Fig. 1b is male (ZZ). *S. margrebowiei* was also judged as ZW, but the data shown in Fig. S1 is again male. I am not sure how they can tell ZW systems by looking at only males. Because the determination of sex chromosome systems is one of the most important parts of this paper, I think that this is serious weakness.

If my reading of the methods is correct (L830-850), they used only coverage data to identify sex-linked genes and annotate them to the Nigon elements. This is highly error-prone. The mapping rates depend on many factors, including degeneration, sequence divergence, gene duplication, and chance effects etc. I suggest they use other methods too.

They often refer to recombination suppression, but I wonder how they can decide whether certain chromosomal regions stop recombination without any linkage map data.

For dosage compensation analysis (Fig 4), which tissues are used? If gonads were used, MSCI will be a confounding factor. It is not clear whether the same tissues are used for different species. Please clarify it.

The authors did a lot of analyses in this paper. It is great. But, I suggest that authors focus their questions on a few important ones. Otherwise, the readers become at a loss what are the main messages of the paper. In L103-L132, you listed a lot of questions, but I think that you can organize these much better.

The novelty of the work is not well explained. Here are several examples.

L167-179: Which data support this claim? Are all these arguments based on previous studies or present studies? It is not clear to me.

L190: Which seven species? It is not clear which are new findings and which are already known before this study. Please clarify the novelty of the work not only in this part but also throughout the paper.

Minor comments

L43: What do you mean by "sexual dimorphism"?

L62: I cannot understand why you use "However" here. Which part is contrary to the previous sentence?

Reviewer #2:

Remarks to the Author:

Review of: "Evolution of sexual systems, sex chromosomes and sex-linked gene transcription in flatworms and roundworms"

Summary: The mechanisms that regulate sexual development can change rapidly during evolution. On rare occasions, the process of determining sex has been altered to depend on sex chromosomes, a change that has major implications for chromosomal composition and evolution. In this manuscript, the authors analyze the genomes of nematodes and flatworms, in order to identify sex chromosomes and determine how they have changed over time. First, they identify sex-linked chromosomal regions for seven male/female species of nematodes. Second, they reconstruct the history of chromosome fusions and that led to the current sex chromosomes in each nematode clade; their work suggests that some of the clades independently evolved the use of sex chromosomes. Third, they further dissect the history of these X chromosomes into strata with different characteristics of recombination, repeat number, etc. Fourth, they study the rate at which they observe the accumulation of female-biased expression on the X chromosome, and the loss of male-biased expression. Their analyses suggest that dosage compensation has been established in these species, but that it is not complete. Finally, they identify orthologs of genes that regulate sex determination in the model nematode *C. elegans*. This last aim is unlikely to explain how sex-determination changes, since we don't know if these genes control sex in these other species. However, the genomic dissection of the structure and evolution of sex chromosomes was very interesting and likely to be of use to many future studies.

Recommendation: This paper is generally well written and addresses an intriguing and important topic that should be of general interest. Although many of the conclusions in the first half are solid and exciting, some results are weak, and the manuscript would have to be revised before acceptance.

Comments:

(1) The authors write as if genes that determine sex in one species are predicted to do so in all others. While this might be true for close relatives, there has been tremendous change in the regulation of sex over larger time scales. For example, the transcription factor TRA-1 specifies female development in *C. elegans*, but its ortholog in *Drosophila* plays no role in sex determination. Thus, the authors need to be very careful in discussing the identification and possible functions of genes orthologous to those that control sex in *C. elegans*.

For example, in lines 604-5 they write: "Within the Z-linked S0 region of *S. mansoni*, we found the ortholog of the *C. elegans* gene mag-1." The identification of mag-1 at this location is intriguing, but in *C. elegans* it appears to be part of a group of genes that functions upstream in the pathway and regulates RNA use. There is no evidence here that it plays any role in sex determination, and the authors should be careful not to overinterpret this finding.

Furthermore, in lines 620-2 they write: "Therefore, the fox-1 paralog may have been integrated into the sex-determination pathway of *S. mansoni* after gonochorism evolved." They must be careful, because there is no evidence that this gene controls sex in *S. mansoni*, which is in a different phylum from nematodes.

Their assumptions also undercut their own findings. For example, on line 695-6 they write: "As expected, *C. elegans* orthologs of feminized genes in *S. mansoni* were enriched for mutant phenotypes related to female reproduction, e.g., egg laying or vulva development." This result is not necessarily expected. They should describe it in more detail because it might be an important new finding.

(2) Along these same lines the authors need to focus on what types of changes have been identified in sex-determination pathways, and whether these changes could be revealed through genomic methods. For example, in lines 778-9 they write: "The latter genes [homologs of *Cel* SD genes] could be involved in divergence of the sex determination pathways that has already been well-documented between *C. elegans* vs. *C. briggsae*." However, the identification of the novel, recently evolved genes she-1 in *C. briggsae* and fog-2 in *C. elegans* suggest that these evolutionary changes depend in large part on genomic innovation, which could not be picked up by the methods in this paper.

(3) The authors introduce flatworms and roundworms as members of two different phyla in the introduction, but the structure of the Figures 1 and 5 obscure this point and might lead the general reader to assume these groups are more closely related than they actually are.

(4) The Introduction needs at least a full paragraph (if not more) on the identification and nature of Nigon elements, to help the general reader follow the detailed discussions of nematode chromosome evolution in the manuscript.

(5) This paper must include a discussion of work from Adrian Streit's lab on sex chromosome diminution in *Strongyloides* and on differential chromatin amplification. They should discuss the results both with respect to their bioinformatic methodology, and to possible mechanisms of evolutionary change.

(6) In Figure 2, the authors use labels as described in lines 329-331: "Nodes inferred to have had XY systems (which we infer to be the ancestral state, see the main text) are indicated with dark blue dots, and nodes with X/O systems with brown dots." Unfortunately, it is hard to distinguish these two dark colors. They should make one of them lighter.

(7) In lines 708-9 the authors write: "Evolution of sex chromosomes involves the primary transition from a hermaphroditic ancestor to a dioecious/gonochoristic species with genetic sex determination." Although this is one reasonable scenario, it could also involve a transition from environmental sex determination.

(8) In lines 769-70 the authors raise this idea, but remain skeptical: "Holocentric chromosomes may also be more prone to fusions or fissions than monocentric chromosomes." I agree with their skepticism and note that if this model were true, one would expect it would not be possible to detect the presence of Nigon groups after the passage of much time.

Minor Changes

line 39 Change "in the two speciose phyla" to "in these two speciose phyla"

lines 87-8 Change "Regardless recombination" to "Regardless of whether recombination"

line 154 Change "to the rest genome," to "to the rest of the genome,"

line 239 Change "chromosome originated in the ancestor" to "chromosome that originated in the ancestor"

line 352 Change ". the" to ". The"

line 370 Change "which has no or much lower" to "which have no or much lower"

line 432 Change "Strata are labelledS0" to "Strata are labelled S0"

line 516 Change "These patterns were the same regardless we included" to "These patterns were the same regardless of whether we included"

line 725 Change "mutation creating females" to "mutation creating males"

line 820 Delete "using Bowtie2 v2.2.981"

line 889 Change "female SNPs densities" to "female SNP densities"

Reviewer #3:

Remarks to the Author:

The ms has looked at the evolution of sexual systems and sex chromosomes in flatworms and roundworms. Wang and colleagues have used publicly available genomes from 54 species. Using male/female coverage ratio (and other proxies), they have identified sex-linked genes in species with sex chromosomes. They present a thorough characterization of sex chromosome and gene expression evolution during the transitions among gonochorism and hermaphroditism in these phyla.

I agree with the authors that there are very few studies addressing the question of the evolution of sex chromosomes in the context of sexual system transitions in animals. I found that their model systems – transition to gonochorism in flatworms and transition to hermaphroditism in roundworms – are very relevant to study this question. They have analysed a lot of data and performed many analyses. Their findings on changes in gene expression in the gonads in Schistosomes (where gonochorism and de novo ZW chromosomes have evolved from a hermaphroditic ancestor) are very interesting. I am thus very positive about this work.

I only have minor changes to suggest:

- p 4, "While gynodioecy (females and hermaphrodites) is regularly observed in plants as the likely intermediate sexual system", perhaps "one of the most likely"? Monoecy is also considered a likely intermediate sexual system (Renner, S. S., & Ricklefs, R. E. (1995). Dioecy and its correlates in the flowering plants. *American journal of botany*, 82(5), 596-606.; Renner, S. S., & Müller, N. A. (2021). Plant sex chromosomes defy evolutionary models of expanding recombination suppression and genetic degeneration. *Nature plants*, 7(4), 392-402).

- p 4, "Androdioecy ... is rare in both plants and animals, and generally represents a transition from gonochorism to hermaphroditism ..., rather than the reverse (2,4)". If I am not wrong, refs 2 and 4 are about gynodioecy in plants. Could you justify briefly that the same conclusions apply to animals?

- p 5, line 65, ")" missing

- p 5, "so-called –turnover events", add some refs.

- p 28, "the X chromosome is expected to accumulate genes with female-biased expression ('feminization') and become depleted for genes with male-biased expression ('demasculinization'), either through expression changes or relocation of individual genes onto other chromosomes, and the opposite ('masculinization') of the Y chromosome." This is true for the dominant mutations. For the recessive ones it is the contrary. Recessive mutations are always exposed in X-hemizygous males compared to only when they are homozygous in females. From the theoretical perspective, masculinisation of the X is also possible (Ellegren, H., & Parsch, J. (2007). The evolution of sex-biased genes and sex-biased gene expression. *Nature Reviews Genetics*, 8(9), 689-698).

- p 41, line 725, "or a mutation creating females (producing an androdioecious population)" replace 'female' by 'male'

- p 49, line 872, remove 1 'between'

- p 49, line 885, can you provide some information on the principle of the change-point analysis?

- Figure 5: I found Figure 5k somewhat confusing. Sexual system of the ancestors of Schistosoma and Nematoda should be different (and not hermaphroditism for both as shown).

REVIEWER #1

Summary: The authors collected publicly available genome sequence data and characterized sex chromosomes of flatworms and roundworms. Although I find some data interesting, I cannot follow the methods or their reasoning in several parts.

1.1: It is not clear how they determined the sex chromosome systems (Fig. 1). According to the methods, they used the coverage data (some examples are shown in Fig. 2b and all data are shown in Fig. S1). I cannot understand how they can determine the sex chromosome systems based on these data. For example, *B. pahangi* is hemephrodite, but there are two peaks. *S. mattheei* was judged as ZW, but the data shown in Fig. 1b is male (ZZ). *S. margrebowiei* was also judged as ZW, but the data shown in Fig. S1 is again male. I am not sure how they can tell ZW systems by looking at only males. Because the determination of sex chromosome systems is one of the most important parts of this paper, I think that this is serious weakness.

R: Concerning these issues, *B. pahangi* is a gonochoristic species, not hermaphroditic, and is closely related to *B. malayi*, which explains its coverage distribution. We infer that the three peaks that are visible in Supplementary Fig. S1 correspond to sequences derived from autosomes, sex chromosomes, and repetitive regions with very low numbers of uniquely mapped reads.

For *S. mattheei* and *S. margrebowiei*, it is correct that we did not identify their sex chromosome type in this work. Our statements are based on the previously reported cytogenetic data (Grossman et al. 1981), and we annotated the chromosomes according to the ZW system for schistosomes. In the revised manuscript, we have added this information in the legend of Figure 1b (lines 220-222): “For some species for which only data for the homogametic sex were available (including *S. mattheei* and *S. margrebowiei*), we labelled their sex chromosome type based on cytogenetic evidence (see Supplementary Table S4)”. We also included the original references in Supplementary Table S4 for the species for which cytogenetic data had been reported (Goldstein 1978; Grossman et al. 1981; Denich and Samoiloff 1984; Spakulova et al. 1994; Špakulová et al. 2001; Post 2005; Nemetschke et al. 2010; Kulkarni et al. 2013; Foster et al. 2020b; Xu et al. 2021).

1.2: If my reading of the methods is correct (L830-850), they used only coverage data to identify sex-linked genes and annotate them to the Nigon elements. This is highly error-prone. The mapping rates depend on many factors, including degeneration, sequence divergence, gene duplication, and chance effects etc. I suggest they use other methods too.

1.3 They often refer to recombination suppression, but I wonder how they can decide whether certain chromosomal regions stop recombination without any linkage map data.

R: The reviewer is correct that various approaches are needed to identify sex-linked regions of different kinds and ages, as reviewed by (Palmer et al. 2019), and we did indeed use both coverage (Supplementary Fig. S2) and SNP (Supplementary Fig. S3) based approaches. Their usefulness and reliability depend on the extent of sequence differentiation between the X/Y or Z/W chromosomes, and the size of the completely sex-linked region. For homomorphic sex chromosomes, GWAS-, Fst-

and SNP-based approaches are suggested, but they require resequencing data for individual males and females, which are presently unavailable. We therefore previously mentioned in line 169-170 that “Physically small or non-differentiated sex-linked regions, however, may not be detectable by our approach”.

Coverage analysis is excellent for sex-linked regions with high divergence, including heteromorphic sex chromosomes, which are present in most nematodes studied here, and diversity analyses (including SNP density) for those with intermediate divergence levels. Our original text (lines 153-160) made clear that the coverage approach was employed to identify the sex chromosomes in species with highly degenerated Y or W chromosomes (Supplementary Fig. S2). Coverage analysis has previously been used to identify the sex chromosomes of nematode and schistosome species, and lines 166-169 now emphasize that “For all 8 species whose sex chromosomes have previously been identified, our coverage pipeline yielded results in agreement with the published ones (Protasio et al. 2012; Foth et al. 2014; Cotton et al. 2016; Hunt et al. 2016; Rödelsperger et al. 2017; Wang et al. 2017; Doyle et al. 2020; Foster et al. 2020a), confirming that this approach is reliable (Supplementary Table S1)”. The results based on SNPs are shown in Supplementary Fig. S3 because they support the coverage pattern, as expected for highly diverged sex-linked regions, since SNP calling is affected by reduced read coverage in the heterogametic sex (it is not included to add new or clearer information). During the revision, we also reorganized the text to explain the principle of coverage-based analysis (lines 153 to 170 at the beginning of the results section).

Regarding the mapping issue raised by the reviewer, we now provide a new Supplementary Table 2 which summarizes the high mapping rates for most species studied here. 39 of the 54 species studied have rates of uniquely mapped reads > 90%. We also explain that we first annotated the Nigon elements based on one-to-one orthologous relationships of genes with those of *C. elegans*, using protein alignments, and then mapped the reads from individual species to their own genome. To avoid possible mapping errors (including cross-mapping between distinct chromosomes or complications due to gene duplication or other artifacts), mapping quality (-q 33) was assessed in each case, and again only uniquely mapped reads were retained. This is explained in line 836-838.

To make our inferences about recombination suppression clearer, we explained the principle of coverage analysis on page 7. The revised text (lines 156-164 onwards) now reads: “Such low coverage results from substitutions accumulated in Y- or W-linked sequences, together with Y- or W-specific transposable element insertions, both of which hinder mapping of sequencing reads to their counterpart X- or Z-linked regions, plus the direct effect on coverage of large deletions (Bachtrog 2013). Coverage analysis, without linkage map data for the sex chromosomes, can thus detect highly divergent sex-linked regions (Palmer et al. 2019), and is widely used, including in birds (Zhou et al. 2014), teleosts (Sardell et al. 2021), reptiles and Lepidoptera (Yoshido et al. 2020). It has also previously been used to detect sex-linked regions in schistosomes (Picard et al. 2018), and to identify the X-linked regions in several nematode species (Hunt et al. 2016; Foster et al. 2020b)”.

1.4 For dosage compensation analysis [sic.] (Fig 4), which tissues are used? If gonads were used, MSCI will be a confounding factor. It is not clear whether the same tissues are used for different species. Please clarify it.

R: We have revised the text (line 548-549) to make this clear: “Comparisons of the transcriptomes from whole adults between fully X-linked *versus* autosomal genes in males (X/A ratios)”. MSCI is indeed a confounding factor, and this has been acknowledged in the revised text (lines 536-538): “Meiotic sex chromosome inactivation (MSCI) in male gonads can also lead to evolution of an underrepresentation of male-biased genes on X chromosomes..”.

1.5 The authors did a lot of analyses in this paper. It is great. But, I suggest that authors focus their questions on a few important ones. Otherwise, the readers become at a loss what are the main messages of the paper. In L103-L132, you listed a lot of questions, but I think that you can organize these much better.

R: We have followed these suggestions, and reorganized and revised the text (lines 97-136).

The novelty of the work is not well explained. Here are several examples.

1.6 L167-179: Which data support this claim? Are all these arguments based on previous studies or present studies? It is not clear to me.

R: These arguments are based on our previous study (Xu et al. 2021). The specific data are shown in Supplementary Fig. S1. We revised the text at line 179-188 to make this clearer. We compared the previously identified S0 region of *S. japonicum* to the sex-linked region of *S. haematobium* identified here, and found that they overlapped. Therefore, we conclude that S0 likely originated in the ancestor of schistosome species.

1.7 L190: Which seven species? It is not clear which are new findings and which are already known before this study. Please clarify the novelty of the work not only in this part but also throughout the paper.

R: We apologize for the confusion. Previously we had marked the species in Fig. 1c without mentioning them in the text; we have now revised the text (lines 192-193): “... (indicated by red arrowheads in Fig. 1c)”. In addition, we previously described 7 species for which sex-linked regions were newly identified. However, we also newly annotated the Nigon elements for 17 species, that was not previously reported. This novel information is now shown in Figure 1c and updated in the text.

Minor comments

1.8 L43: What do you mean by "sexual dimorphism"?

R: Sexual dimorphism is a commonly used term; it refers to differences between the sexes of the same species, including morphological and behavioral traits, and also gene expression. We have now

explained the term and included a reference (Hedrick and Temeles 1989) at its first appearance in the text in line 36.

1.9 L62: I cannot understand why you use "However" here. Which part is contrary to the previous sentence?

R: We restructured and improved this sentence in line 58.

REVIEWER #2:

Summary: The mechanisms that regulate sexual development can change rapidly during evolution. On rare occasions, the process of determining sex has been altered to depend on sex chromosomes, a change that has major implications for chromosomal composition and evolution. In this manuscript, the authors analyze the genomes of nematodes and flatworms, in order to identify sex chromosomes and determine how they have changed over time. First, they identify sex-linked chromosomal regions for seven male/female species of nematodes. Second, they reconstruct the history of chromosome fusions and that led to the current sex chromosomes in each nematode clade; their work suggests that some of the clades independently evolved the use of sex chromosomes. Third, they further dissect the history of these X chromosomes into strata with different characteristics of recombination, repeat number, etc. Fourth, they study the rate at which they observe the accumulation of female-biased expression on the X chromosome, and the loss of male-biased expression. Their analyses suggest that dosage compensation has been established in these species, but that it is not complete. Finally, they identify orthologs of genes that regulate sex determination in the model nematode *C. elegans*. This last aim is unlikely to explain how sex-determination changes, since we don't know if these genes control sex in these other species. However, the genomic dissection of the structure and evolution of sex chromosomes was very interesting and likely to be of use to many future studies.

Recommendation: This paper is generally well written and addresses an intriguing and important topic that should be of general interest. Although many of the conclusions in the first half are solid and exciting, some results are weak, and the manuscript would have to be revised before acceptance.

R: Thank you for this concise summary and for the positive comments on our work. In the following, we address the issues raised.

Comments:

2.1 The authors write as if genes that determine sex in one species are predicted to do so in all others. While this might be true for close relatives, there has been tremendous change in the regulation of sex over larger time scales. For example, the transcription factor TRA-1 specifies female development in *C. elegans*, but its ortholog in *Drosophila* plays no role in sex determination. Thus, the authors need to be very careful in discussing the identification and possible functions of genes orthologous to those that control sex in *C. elegans*.

R: We apologize if our writing caused confusion. It is correct that sex determination has changed over evolutionary time, including turnovers that result in different systems in different species. We have clarified (lines 633-642) that “Our inferences of genes potentially participating in the sex-determination pathways of flatworms and nematodes other than *C. elegans* used the well characterized *C. elegans* genes as a reference. However, it is known that sex-determining pathway genes can undergo rapid turnovers between even related species (Haag 2005; Pan et al. 2016), therefore orthologs of *C. elegans* genes in other nematode species, and particularly the deeply diverged flatworms do not necessarily have a sex-determining function and need to be functionally validated in the focal species in future. Below we mainly focused our description of certain genes that have functional evidence in both schistosomes and *C. elegans*.”

2.2 For example, in lines 604-5 they write: “Within the Z-linked S0 region of *S. mansoni*, we found the ortholog of the *C. elegans* gene *mag-1*.” The identification of *mag-1* at this location is intriguing, but in *C. elegans* it appears to be part of a group of genes that functions upstream in the pathway and regulates RNA use. There is no evidence here that it plays any role in sex determination, and the authors should be careful not to overinterpret this finding.

R: We revised the text to describe *mag-1* more precisely (lines 653-662) that “Within the Z-linked S0 region of *S. mansoni*, we found the ortholog of the *C. elegans* gene *mag-1*, whose knockdown in *C. elegans* causes the hermaphrodites to produce sperm instead of oocytes (Li et al. 2000); in *S. japonicum* its knockdown causes apparent cell proliferation in testicular lobes (Zhao et al. 2008). Functional experiments in *Drosophila* (Micklem et al. 1997) have indicated an important and evolutionarily conserved role of *mag-1* during oogenesis. If gonochorism originated in a non-gonochoristic schistosome ancestor via two mutations, a possible scenario might be that recessive female-suppressing mutations have first affected *mag-1*'s likely feminizing function in the germline, leading to a transition from hermaphroditism to androdioecy as the first step of evolution of gonochorism (Charlesworth and Charlesworth 1978)”.

2.3 Furthermore, in lines 620-2 they write: “Therefore, the *fox-1* paralog may have been integrated into the sex-determination pathway of *S. mansoni* after gonochorism evolved.” They must be careful, because there is no evidence that this gene controls sex in *S. mansoni*, which is in a different phylum from nematodes.

R: We removed “another possible candidate sex-determining gene” when describing *fox-1*, and now state in lines 670-671 that the “Therefore, the *fox-1* paralog of *S. mansoni* may not have a sex-determining function.”.

2.4 Their assumptions also undercut their own findings. For example, on line 695-6 they write: “As expected, *C. elegans* orthologs of feminized genes in *S. mansoni* were enriched for mutant phenotypes related to female reproduction, e.g., egg laying or vulva development.” This result is not

necessarily expected. They should describe it in more detail because it might be an important new finding.

R: We have performed new enrichment analyses of the mutant phenotypes for *C. elegans* orthologs of feminized and masculinized genes defined in *S. mansoni* relative to its hermaphroditic relatives, and list all the results in new Supplementary Tables 11 and 12. As mentioned in the main text, feminized or masculinized genes were defined by comparing the gonad expression levels between *S. mansoni* vs. its hermaphroditic related species. For example, a gene showing at least 2-fold higher expression levels in *S. mansoni*'s ovary relative to *C. sinensis* is defined as a 'feminized' genes. Apart from female-related mutant phenotypes, such as 'oogenesis variant', 'vulva morphology variant' and 'hermaphrodite fertility reduced', orthologous feminized genes of *S. mansoni* are also enriched for *C. elegans* mutant phenotypes impacting early embryonic or germ cell development (e.g., 'embryonic lethal', 'development timing variant' and 'meiosis variant'). In contrast, male-related phenotypes (e.g., 'male mating variant' and 'male response to hermaphrodite variant') are found only in masculinized, but not in feminized genes. We removed "As expected", and revised lines 740-746 according in response to this comment.

2.5 Along these same lines the authors need to focus on what types of changes have been identified in sex-determination pathways, and whether these changes could be revealed through genomic methods. For example, in lines 778-9 they write: "The latter genes [homologs of *Cel* SD genes] could be involved in divergence of the sex determination pathways that has already been well-documented between *C. elegans* vs. *C. briggsae*." However, the identification of the novel, recently evolved genes *she-1* in *C. briggsae* and *fog-2* in *C. elegans* suggest that these evolutionary changes depend in large part on genomic innovation, which could not be picked up by the methods in this paper.

R: We have revised lines 814-825 to read: "These genes could be involved in divergence of the sex-determination pathways, as has already been documented between *C. elegans* vs. *C. briggsae* (Stothard and Pilgrim 2003). Functional verification in other nematode species is needed in the future. The present study cannot identify further changes that may have occurred after lineage-specific duplications of these genes, and possible changes also need to be studied further. For example, *gld-1* was independently recruited into the sex-determination pathways of *C. elegans* and *C. briggsae*; in *C. elegans* it acts to promote spermatogenesis, but it promotes oogenesis in *C. briggsae* (Beadell et al. 2011). Its co-factor *fog-2* evolved by a duplication and acquisition of a new GLD-1-binding domain in *C. elegans* (Clifford et al. 2000; Haag et al. 2018). The newly annotated candidate sex-determining genes will be a useful resource for future studies of parasite control through interfering with their sexual life cycles".

2.6 The authors introduce flatworms and roundworms as members of two different phyla in the introduction, but the structure of the Figures 1 and 5 obscure this point and might lead the general reader to assume these groups are more closely related than they actually are.

R: We revised the text (lines 97-101) that “Here we investigate representative species of two speciose and widely distributed phyla, Platyhelminthes (flatworms including the classes Trematoda and Cestoda) and Nematoda (Nemathelminthes, the roundworms or nematodes), to reconstruct the origin and evolution of sex chromosomes. These phyla diverged about 700 million years ago (MYA) (Parfrey et al. 2011)”. We separated the two phyla in Fig.1 by adding more space between them, and revised Fig 5c by removing the Schistosoma branch, so that the reader is not misled into thinking they are closely related.

2.7 The Introduction needs at least a full paragraph (if not more) on the identification and nature of Nigon elements, to help the general reader follow the detailed discussions of nematode chromosome evolution in the manuscript.

R: Thank you for this suggestion. We have added a paragraph (lines 113-123): “These seven chromosome elements were first proposed after comparing the genomes of rhabditid nematodes (Rhabditida order, including clades III, IV, and V, see below) (Tandonnet et al. 2019; Gonzalez de la Rosa et al. 2021), most of which have a karyotype with $n=6$. Orthologous genes are found on homologous chromosomes across different species, defining the Nigon elements, although gene orders within elements differ because of rearrangements, and different elements may undergo fusions or fissions. The five *C. elegans* autosome pairs correspond to elements NA to NE, and individual orthologous genes are rarely found on different elements. However, the NN and NX elements that together constitute the *C. elegans* X (see also below) are exceptions. We now study the sex chromosomes of species from all major nematode clades, and assign their genes to the component Nigon elements, based on orthology with *C. elegans* genes”.

2.8 This paper must include a discussion of work from Adrian Streit’s lab on sex chromosome diminution in *Strongyloides* and on differential chromatin amplification. They should discuss the results both with respect to their bioinformatic methodology, and to possible mechanisms of evolutionary change.

R: We apologize for this omission, and thank you for raising this. Our revised manuscript mentions that *Strongyloides* species have environmental sex determination, rather than genetic sex determination as in many other nematodes. We have now added asterisks on the *Strongyloides* node in Figure 1c, and described its sex determination mechanism in the legend and in the text (lines 225-229) that “We marked the parasitic *Strongyloides* clade here and in Figure 2d below with an asterisk to indicate that this group (including *S. ratti* and *S. papillosus*) have environmental sex determination. Males of this group develop in response to the host immune responses, which triggers loss of X-linked sequences during oocyte mitosis (Nemetschke et al. 2010)”.

The original work by Streit’s group (Nemetschke et al. 2010) did not adopt a bioinformatic method but used ~ 50 EST markers to test genes for hemizyosity in males, which allowed them to infer the X-linked region, and showed that it corresponds to the region that undergoes chromatin diminution in *S. papillosus* whose X is fused to an autosome. A subsequent study of *Strongyloides*

(Hunt et al. 2016) used the same approach as ours, based on a read coverage difference between sexes, to infer the sequences that are X-linked. We have revised the text (lines 313-326) to make this clear.

2.9 In Figure 2, the authors use labels as described in lines 329-331: “Nodes inferred to have had XY systems (which we infer to be the ancestral state, see the main text) are indicated with dark blue dots, and nodes with X/O systems with brown dots.” Unfortunately, it is hard to distinguish these two dark colors. They should make one of them lighter.

R: We apologize, and have now made the brown dots in Figures 2 and 3 a lighter colour.

2.10 In lines 708-9 the authors write: “Evolution of sex chromosomes involves the primary transition from a hermaphroditic ancestor to a dioecious/gonochoristic species with genetic sex determination.” Although this is one reasonable scenario, it could also involve a transition from environmental sex determination.

R: We have revised (lines 757-759) that “The evolution of sex chromosomes in some taxa involves the primary transition from a hermaphroditic or from ESD to a dioecious/gonochoristic species with genetic sex determination or an ancestor with environmental sex determination..”.

2.11 In lines 769-70 the authors raise this idea, but remain skeptical: “Holocentric chromosomes may also be more prone to fusions or fissions than monocentric chromosomes.” I agree with their skepticism and note that if this model were true, one would expect it would not be possible to detect the presence of Nigon groups after the passage of much time.

R: Thank you for raising this issue. We have revised the text accordingly (lines 797-803) that “However, the study of fusions between the *C. elegans* X chromosome and chromosome IV (Henzel et al. 2011) suggested that crossovers may be re-positioned away from the fusion junction, creating a new chromosome with two arm regions (whereas the two participating chromosomes each contained two arm regions). In the fused chromosome, a potentially large former arm region close to the fusion point may thus have greatly reduced recombination, and if the fusion involves the X chromosome, this will occur specifically in males.”.

2.12 Minor Changes

line 39 Change “in the two speciose phyla” to “in these two speciose phyla”

R: Revised (see line 97).

lines 87-8 Change “Regardless recombination” to “Regardless of whether recombination”

R: Revised (see line 555).

line 154 Change “to the rest genome,” to “to the rest of the genome,”

R: Revised the sentence.

line 239 Change “chromosome originated in the ancestor” to “chromosome that originated in the ancestor”

R: Revised the sentence.

line 352 Change “. the” to “. The”

R: Revised (see line 376).

line 370 Change “which has no or much lower” to “which have no or much lower”

R: Revised the sentence).

line 432 Change “Strata are labelledS0” to “Strata are labelled S0”

R: Revised (see line 472).

line 516 Change “These patterns were the same regardless we included” to “These patterns were the same regardless of whether we included”

R: Revised (see line 555).

line 725 Change “mutation creating females” to “mutation creating males”

R: Revised (see line 769).

line 820 Delete “using Bowtie2 v2.2.981”

R: Revised (see line 837).

line 889 Change “female SNPs densities” to “female SNP densities”

R: Revised (see line 937).

REVIEWER #3:

3.1 Summary: The ms has looked at the evolution of sexual systems and sex chromosomes in flatworms and roundworms. Wang and colleagues have used publicly available genomes from 54 species. Using male/female coverage ratio (and other proxies), they have identified sex-linked genes in species with sex chromosomes. They present a thorough characterization of sex chromosome and gene expression evolution during the transitions among gonochorism and hermaphroditism in these phyla.

I agree with the authors that there are very few studies addressing the question of the evolution of sex chromosomes in the context of sexual system transitions in animals. I found that their model systems – transition to gonochorism in flatworms and transition to hermaphroditism in roundworms – are very relevant to study this question. They have analysed a lot of data and performed many analyses. Their findings on changes in gene expression in the gonads in Schistosomes (where gonochorism and de novo ZW chromosomes have evolved from a hermaphroditic ancestor) are very interesting. I am thus very positive about this work.

I only have minor changes to suggest:

R: Thank you for these positive comments.

3.2 - p 4, “While gynodioecy (females and hermaphrodites) is regularly observed in plants as the likely intermediate sexual system”, perhaps “one of the most likely”? Monoecy is also considered a likely intermediate sexual system (Renner, S. S., & Ricklefs, R. E. (1995). Dioecy and its correlates in the flowering plants. *American journal of botany*, 82(5), 596-606.; Renner & Müller (2021). *Plant sex*

chromosomes defy evolutionary models of expanding recombination suppression and genetic degeneration. *Nature plants*, 7(4), 392-402).

R: We revised the text (lines 39-41), and also added the suggested citations as suggested.

3.3 - p 4, “Androdioecy ... is rare in both plants and animals, and generally represents a transition from gonochorism to hermaphroditism ..., rather than the reverse (2,4)” . If I am not wrong, refs 2 and 4 are about gynodioecy in plants. Could you justify briefly that the same conclusions apply to animals?

R: Thanks for pointing this out. We have now added the citation (Weeks 2012) that specifically addresses this issue in animals (line 46).

3.4 - p 5, line 65, “)” missing

R: We could not find this error; both “(” and “)” seem to be present.

3.5 - p 5, “so-called —turnover events”, add some refs.

R: We have added “turnover events (Vicoso 2019; Pan et al. 2021)” citations on line 71.

3.6 - p 28, “the X chromosome is expected to accumulate genes with female-biased expression (‘feminization’) and become depleted for genes with male-biased expression (‘demasculinization’), either through expression changes or relocation of individual genes onto other chromosomes, and the opposite (‘masculinization’) of the Y chromosome.” This is true for the dominant mutations. For the recessive ones it is the contrary. Recessive mutations are always exposed in X-hemizygous males compared to only when they are homozygous in females. From the theoretical perspective, masculinisation of the X is also possible (Ellegren, H., & Parsch, J. (2007). The evolution of sex-biased genes and sex-biased gene expression. *Nature Reviews Genetics*, 8(9), 689-698).

R: We revised (lines 539 and 542) that “the X chromosome is also expected to accumulate genes with female-biased expression (‘feminization’) and become depleted for genes with male-biased expression (‘demasculinization’) if the mutations are expressed in heterozygotes (wholly or partially dominant)..”.

3.7 - p 41, line 725, “or a mutation creating females (producing an androdioecious population)” replace ‘female’ by ‘male’

R: Revised (see line 769).

3.8 - p 49, line 872, remove 1 ‘between’

R: Revised (see line 918).

3.9- p 49, line 885, can you provide some information on the principle of the change-point analysis?

R: A change-point analysis detects the data point showing a significant shift of mean values between two consecutive datasets, which could be time-series data, or in this work, certain values of many consecutive genomic windows. We have added this information to the passage (lines 931-933).

3.10 - Figure 5: I found Figure 5k somewhat confusing. Sexual system of the ancestors of Schistosoma and Nematoda should be different (and not hermaphroditism for both as shown).

R: Thank you for pointing out this error. We have included separate hermaphroditic ancestors for the two lineages.

References added:

- Bachtrog D. 2013. Y-chromosome evolution: emerging insights into processes of Y-chromosome degeneration. *Nat Rev Genet* **14**: 113-124.
- Beadell AV, Liu Q, Johnson DM, Haag ES. 2011. Independent recruitments of a translational regulator in the evolution of self-fertile nematodes. *Proc Natl Acad Sci USA* **108**: 19672-19677.
- Charlesworth B, Charlesworth D. 1978. A model for the evolution of dioecy and gynodioecy. *Am Nat* **112**: 975-997.
- Clifford R, Lee M-H, Nayak S, Ohmachi M, Giorgini F, Schedl T. 2000. FOG-2, a novel F-box containing protein, associates with the GLD-1 RNA binding protein and directs male sex determination in the *C. elegans* hermaphrodite germline. *Development* **127**: 5265-5276.
- Cotton JA, Bennuru S, Grote A, Harsha B, Tracey A, Beech R, Doyle SR, Dunn M, Hotopp JCD, Holroyd N et al. 2016. The genome of *Onchocerca volvulus*, agent of river blindness. *Nat Microbiol* **2**: 16216.
- Denich KT, Samoiloff MR. 1984. Estimation of mutation rates induced by large doses of gamma, proton and neutron irradiation of the X-chromosome of the nematode *Panagrellus redivivus*. *Mutation Research Letters* **140**: 103-106.
- Doyle SR, Tracey A, Laing R, Holroyd N, Bartley D, Bazant W, Beasley H, Beech R, Britton C, Brooks K. 2020. Genomic and transcriptomic variation defines the chromosome-scale assembly of *Haemonchus contortus*, a model gastrointestinal worm. *Commun Biol* **3**: 1-16.
- Foster JM, Grote A, Mattick J, Tracey A, Tsai Y-C, Chung M, Cotton JA, Clark TA, Geber A, Holroyd N et al. 2020a. Sex chromosome evolution in parasitic nematodes of humans. *Nat Commun* **11**: 1964.
- Foster JM, Grote A, Mattick J, Tracey A, Tsai YC, Chung M, Cotton JA, Clark TA, Geber A, Holroyd N et al. 2020b. Sex chromosome evolution in parasitic nematodes of humans. *Nat Commun* **11**: 1964.
- Foth BJ, Tsai IJ, Reid AJ, Bancroft AJ, Nichol S, Tracey A, Holroyd N, Cotton JA, Stanley EJ, Zarowiecki M. 2014. Whipworm genome and dual-species transcriptome analyses provide molecular insights into an intimate host-parasite interaction. *Nat Genet* **46**: 693-700.
- Goldstein P. 1978. Ultrastructural analysis of sex determination in *Ascaris lumbricoides* var. *suum*. *Chromosoma* **66**: 59-69.
- Gonzalez de la Rosa PM, Thomson M, Trivedi U, Tracey A, Tandonnet S, Blaxter M. 2021. A telomere-to-telomere assembly of *Oscheius tipulae* and the evolution of rhabditid nematode chromosomes. *G3 (Bethesda)* **11**: 1-17.
- Grossman AI, Short RB, Cain GD. 1981. Karyotype evolution and sex chromosome differentiation in Schistosomes (Trematoda, Schistosomatidae). *Chromosoma* **84**: 413-430.
- Haag ES. 2005. The evolution of nematode sex determination: *C. elegans* as a reference point for comparative biology. *WormBook*: 1.
- Haag ES, Fitch DHA, Delattre M. 2018. From "the Worm" to "the Worms" and Back Again: The Evolutionary Developmental Biology of Nematodes. *Genetics* **210**: 397-433.

- Hedrick AV, Temeles EJ. 1989. The evolution of sexual dimorphism in animals: hypotheses and tests. *Trends in ecology & evolution* **4**: 136-138.
- Henzel JV, Nabeshima K, Schvarzstein M, Turner BE, Villeneuve AM, Hillers KJ. 2011. An asymmetric chromosome pair undergoes synaptic adjustment and crossover redistribution during *Caenorhabditis elegans* meiosis: implications for sex chromosome evolution. *Genetics* **187**: 685-699.
- Hunt VL, Tsai IJ, Coghlan A, Reid AJ, Holroyd N, Foth BJ, Tracey A, Cotton JA, Stanley EJ, Beasley H. 2016. The genomic basis of parasitism in the *Strongyloides* clade of nematodes. *Nat Genet* **48**: 299-307.
- Kulkarni A, Dyka A, Nemetschke L, Grant WN, Streit A. 2013. Parastrongyloides trichosuri suggests that XX/XO sex determination is ancestral in *Strongyloididae* (Nematoda). *Parasitology* **140**: 1822-1830.
- Li W, Boswell R, Wood WB. 2000. *mag-1*, a homolog of *Drosophila mago nashi*, regulates hermaphrodite germline sex determination in *Caenorhabditis elegans*. *Dev Biol* **218**: 172-182.
- Micklem DR, Dasgupta R, Elliott H, Gergely F, Davidson C, Brand A, González-Reyes A, St Johnston D. 1997. The *mago nashi* gene is required for the polarisation of the oocyte and the formation of perpendicular axes in *Drosophila*. *Curr Biol* **7**: 468-478.
- Nemetschke L, Eberhardt AG, Hertzberg H, Streit A. 2010. Genetics, chromatin diminution, and sex chromosome evolution in the parasitic nematode genus *Strongyloides*. *Curr Biol* **20**: 1687-1696.
- Palmer DH, Rogers TF, Dean R, Wright AE. 2019. How to identify sex chromosomes and their turnover. *Mol Ecol* **28**: 4709-4724.
- Pan Q, Anderson J, Bertho S, Herpin A, Wilson C, Postlethwait JH, Schartl M, Guiguen Y. 2016. Vertebrate sex-determining genes play musical chairs. *C R Biol* **339**: 258-262.
- Pan Q, Kay T, Depincé A, Adolphi M, Schartl M, Guiguen Y, Herpin A. 2021. Evolution of master sex determiners: TGF- β signalling pathways at regulatory crossroads. *Philos Trans R Soc Lond B Biol Sci* **376**: 20200091.
- Parfrey LW, Lahr DJ, Knoll AH, Katz LA. 2011. Estimating the timing of early eukaryotic diversification with multigene molecular clocks. *Proc Natl Acad Sci USA* **108**: 13624-13629.
- Picard MA, Cosseau C, Ferre S, Quack T, Grevelding CG, Coute Y, Vicoso B. 2018. Evolution of gene dosage on the Z-chromosome of schistosome parasites. *eLife* **7**: e35684.
- Post R. 2005. The chromosomes of the Filariae. *Filaria J* **4**: 10.
- Protasio AV, Tsai IJ, Babbage A, Nichol S, Hunt M, Aslett MA, De Silva N, Velarde GS, Anderson TJ, Clark RC. 2012. A systematically improved high quality genome and transcriptome of the human blood fluke *Schistosoma mansoni*. *PLoS Negl Trop Dis* **6**: e1455.
- Rödelsperger C, Meyer JM, Prabh N, Lanz C, Bemm F, Sommer RJ. 2017. Single-molecule sequencing reveals the chromosome-scale genomic architecture of the nematode model organism *Pristionchus pacificus*. *Cell reports* **21**: 834-844.
- Sardell JM, Josephson MP, Dalziel AC, Peichel CL, Kirkpatrick M. 2021. Heterogeneous histories of recombination suppression on stickleback sex chromosomes. *Mol Biol Evol* **38**: 4403-4418.
- Špakulová M, Horák P, Dvořák J. 2001. The karyotype of *Trichobilharzia regenti* Horák, Kolářová et Dvořák, 1998 (Digenea: Schistosomatidae), a nasal avian schistome in Central Europe. *Parasitology Research* **87**: 479-483.
- Spakulova M, Kralova I, Cutillas C. 1994. Studies on the karyotype and gametogenesis in *Trichuris muris*. *J Helminthol* **68**: 67-72.
- Stothard P, Pilgrim D. 2003. Sex-determination gene and pathway evolution in nematodes. *Bioessays* **25**: 221-231.
- Tandonnet S, Koutsovoulos GD, Adams S, Cloarec D, Parihar M, Blaxter ML, Pires-daSilva A. 2019. Chromosome-wide evolution and sex determination in the three-sexed nematode *Auanema rhodensis*. *G3 (Bethesda)* **9**: 1211-1230.
- Vicoso B. 2019. Molecular and evolutionary dynamics of animal sex-chromosome turnover. *Nat Ecol Evol* **3**: 1632-1641.

- Wang J, Gao S, Mostovoy Y, Kang Y, Zagoskin M, Sun Y, Zhang B, White LK, Easton A, Nutman TB. 2017. Comparative genome analysis of programmed DNA elimination in nematodes. *Genome Res* **27**: 2001-2014.
- Weeks SC. 2012. The role of androdioecy and gynodioecy in mediating evolutionary transitions between dioecy and hermaphroditism in the Animalia. *Evolution* **66**: 3670-3686.
- Xu X, Wang Y, Guan Q, Guo G, Yu X, Dai Y, Liu Y, Wei G, Wang C, He X et al. 2021. Chromosome-level genome assembly reveals female-biased genes for sex determination and differentiation in the human blood fluke *Schistosoma japonicum*. *Authorea* doi:10.22541/au.163632841.12699383/v1.
- Yoshido A, Šichová J, Pospíšilová K, Nguyen P, Voleníková A, Šafář J, Provazník J, Vila R, Marec F. 2020. Evolution of multiple sex-chromosomes associated with dynamic genome reshuffling in Leptidea wood-white butterflies. *Heredity* **125**: 138-154.
- Zhao Z-R, Lei L, Liu M, Zhu S-C, Ren C-P, Wang X-N, Shen J-J. 2008. *Schistosoma japonicum*: inhibition of Mago nashi gene expression by shRNA-mediated RNA interference. *Exp Parasitol* **119**: 379-384.
- Zhou Q, Zhang J, Bachtrog D, An N, Huang Q, Jarvis ED, Gilbert MT, Zhang G. 2014. Complex evolutionary trajectories of sex chromosomes across bird taxa. *Science* **346**: 1246338.

Reviewers' Comments:

Reviewer #1:

Remarks to the Author:

This paper has been greatly improved following suggestions made by the reviewers. I have only minor comments.

L27-28 "have stopped recombining": This is overclaim. As I commented previously, the authors did not directly measure the recombination rate. So, I suggest that they should say, for example, "show genomic signatures of recombination suppression".

L31: I think that "on sex chromosomes" is necessary after "gonad gene expression".

L36: It would be better to say "Dioecy (separate sexes)" or "Having two sexes (males and females)" rather than "sexual dimorphism". They cite the Hedrick & Temeles paper here, but this paper is on sex difference in size and morphology. I do not think that the present manuscript is a paper on sex difference in size and morphology.

L46-47 "transitions from hemaphroditic ancestors to androdioecy may be associated with selection for sexual dimorphism in size": Is this true? They cite a Weeks et al paper here, but this paper is on the transition from dioecy to hemarphrodite.

L136-137: "these parasitic worms that cause diseases in humans and other animals": This part needs citations.

L224: To clarify that these cytogenetic data are based on previous studies, I suggest that they say "based on previously reported cytogenetic data".

Fig. 1C: In the previous manuscript, *A. viteae* was shown as hermaphrodite and *S. muris* was classified as XY or XO. But, in the revised paper, these were changed into XY or XO and hermaphrodite, respectively. Why? Which is correct?

Reviewer #2:

Remarks to the Author:

Review of: "Evolution of sexual systems, sex chromosomes and sex-linked gene transcription in flatworms and roundworms"

Summary: The mechanisms that regulate sexual development can change rapidly during evolution. On rare occasions, the process of determining sex has been altered to depend on sex chromosomes, a change that has major implications for chromosomal composition and evolution. In this manuscript, the authors analyze the genomes of nematodes and flatworms, in order to identify sex chromosomes and determine how they have changed over time. First, they identify sex-linked chromosomal regions for seven male/female species of nematodes. Second, they reconstruct the history of chromosome fusions and that led to the current sex chromosomes in each nematode clade; their work suggests that some of the clades independently evolved the use of sex chromosomes. Third, they further dissect the history of these X chromosomes into strata with different characteristics of recombination, repeat number, etc. Fourth, they study the rate at which they observe the accumulation of female-biased expression on the X chromosome, and the loss of male-biased expression. Their analyses suggest that dosage compensation has been established in these species, but that it is not complete. Finally, they identify orthologs of genes that regulate sex determination in the model nematode *C. elegans*. This last aim is unlikely to explain how sex-determination changes, since we don't know if these genes control sex in these other species. However, the genomic dissection of the structure and

evolution of sex chromosomes was very interesting and likely to be of use to many future studies.

Recommendation: This paper is well written and addresses an intriguing and important topic of general interest. The authors have revised the text to address my initial concerns, and I strongly recommend publication of this detailed and important work. Below I present some comments to improve the final version, but I think they should be simple to address.

Comments

(1) In lines 569-570, the authors write: "This was mainly due to lower genome-wide transcription levels in females, including for autosomal genes..."

How the genome-wide transcriptional levels are being normalized is very important. Is this on a per animal basis, or a per cell basis? Are there basic facts about each sex and what size it reaches that might be relevant here? This section should be expanded to explain the critical background assumptions to the reader.

(2) In lines 572-4, the authors write: "The transcription levels of autosomal genes in male *O. volvulus* are similar to those of their orthologs in *B. malayi*"

Here also it is critical to explain the method for comparing transcription levels between species, which is potentially complex.

(3) Line 45 would be a good place to add a paragraph break, to help the general reader consider each complex evolutionary transition separately. Furthermore, if the authors include a potential selective cause for the transition from hermaphroditism to androdioecy, they should also include one for the transition from gonochorism to androdioecy.

(4) In lines 686-7 the authors write: "Lineage-specific duplication of *C. elegans* sex determining genes across the phylogeny of nematodes" and in lines 708-11 they write: "Duplication of orthologs of genes involved in the sex-determination pathway of *C. elegans* also seems to have occurred in other nematodes, as we found lineage-specific duplications of sex-determination pathway genes on different Nigon elements"

Is it possible that some of these events are not lineage-specific duplications, but rather allelic variants in male/female populations? This concern should be addressed in the text.

(5) In Figure 5k, the authors should revise the nematode portion of the model. (a) The initial part showing hermaphrodites evolving to make a male/female species is largely outside of the scope of this paper and it might be better to eliminate it so as to focus on a male/female species evolving into a male/hermaphrodite one. (b) In this second step, the *mss* genes are certainly very important, but *fog-2* and *she-1* should also be included, as they help control germ cell sex. In addition the authors should consider mentioning *try-5*, *spe-8* and other sperm activation genes.

(6) In line 684 the authors write: "the candidate sex-determining genes *U2AF2*, *fox-1*"

I can't see the arrow for *fox-1*

Minor comments

line 89 Change "an ancestral sex chromosomes." to "an ancestral sex chromosome."

line 134 Change "for future discovering the sex determining" to "for the future discovery of the sex-determining"

line 164 Change "has also previously been used to detected" to "has also previously been used to detect"

line 187 Change "and presence" to "and the presence"

line 204 Change "due to presence" to "due to the presence"

line 264 Change "creating a neo-sex chromosomes" to "creating a neo-sex chromosome"

line 334 Change "for a long evolutionary time" to "for such a long evolutionary time"

line 366 Change "elements that forms the X" to "elements that form the X"

line 403 Change "specifically in male" to "specifically in males"

line 576 Fix "as suggested in Ref."

line 648 Change "in the focal species in future." to "in the focal species in the future."

line 648 Change "our description of" to "our description on"

line 669 Change "as the first step of evolution" to "as the first step in the evolution"

line 701 Change "model that for" to "model for"

REVIEWER #1

Summary: This paper has been greatly improved following suggestions made by the reviewers. I have only minor comments.

R: Thank you!

L27-28 "have stopped recombining": This is overclaim. As I commented previously, the authors did not directly measure the recombination rate. So, I suggest that they should say, for example, "show genomic signatures of recombination suppression".

R: Revised (see line 30).

L31: I think that "on sex chromosomes" is necessary after "gonad gene expression".

R: Actually, gonad-expressed genes that show signatures of masculinization and defeminization are distributed on both autosomes and sex chromosomes of *Schistosoma masoni*, therefore we did not add "on sex chromosomes" here, see line 33.

L36: It would be better to say "Dioecy (separate sexes)" or "Having two sexes (males and females)" rather than "sexual dimorphism". They cite the Hedrick & Temeles paper here, but this paper is on sex difference in size and morphology. I do not think that the present manuscript is a paper on sex difference in size and morphology.

R: Revised (see line 40-42). We also removed Hedrick & Hemeles citation.

L46-47 "transitions from hemaphroditic ancestors to androdioecy may be associated with selection for sexual dimorphism in size": Is this true? They cite a Weeks et al paper here, but this paper is on the transition from dioecy to hemarphrodite.

R: Sorry for this mistake, the correct reference should be:

"S. C. Weeks, The role of androdioecy and gynodioecy in mediating evolutionary transitions between dioecy and hermaphroditism in the Animalia," *Evolution*, vol. 66, no. 12, pp. 3670-3686, 2012." And we have corrected this, see line 52.

L136-137: "these parasitic worms that cause diseases in humans and other animals": This part needs citations.

R: Thank you for pointing this out, we added two citations at line 143:

G.B.D. 2015 Disease and Injury Incidence and Prevalence Collaborators. Global, regional, and national incidence, prevalence, and years lived with disability for 310 diseases and injuries, 1990-2015: a systematic analysis for the Global Burden of Disease Study 2015. *Lancet* 388, 1545–1602 (2016).

Charlier, J., van der Voort, M., Kenyon, F., Skuce, P. & Vercruyse, J. Chasing helminths and their economic impact on farmed ruminants. *Trends Parasitol.* 30, 361–367 (2014).

See line 141.

L224: To clarify that these cytogenetic data are based on previous studies, I suggest that they say "based on previously reported cytogenetic data".

R: Revised (see line 1188-1189).

Fig. 1C: In the previous manuscript, *A. viteae* was shown as hermaphrodite and *S. muris* was classified as XY or XO. But, in the revised paper, these were changed into XY or XO and hermaphrodite, respectively. Why? Which is correct?

R: The latest revised version is correct that *A. viteae* has a XX/XO sex system (Post 2005); while *S. muris*' sex chromosome system is unknown, because there was no previous cytological study, and we cannot determine in this work based on available data either. In the earliest version of ms, there were mistakes labeling these two species, we have now corrected them in the latest version.

REVIEWER #2:

Summary: The mechanisms that regulate sexual development can change rapidly during evolution. On rare occasions, the process of determining sex has been altered to depend on sex chromosomes, a change that has major implications for chromosomal composition and evolution. In this manuscript, the authors analyze the genomes of nematodes and flatworms, in order to identify sex chromosomes and determine how they have changed over time. First, they identify sex-linked chromosomal regions for seven male/female species of nematodes. Second, they reconstruct the history of chromosome fusions and that led to the current sex chromosomes in each nematode clade; their work suggests that some of the clades independently evolved the use of sex chromosomes. Third, they further dissect the history of these X chromosomes into strata with different characteristics of recombination, repeat number, etc. Fourth, they study the rate at which they observe the accumulation of female-biased expression on the X chromosome, and the loss of male-biased expression. Their analyses suggest that dosage compensation has been established in these species, but that it is not complete. Finally, they identify orthologs of genes that regulate sex determination in the model nematode *C. elegans*. This last aim is unlikely to explain how sex-determination changes, since we don't know if these genes control sex in these other species. However, the genomic dissection of the structure and evolution of sex chromosomes was very interesting and likely to be of use to many future studies.

Recommendation: This paper is generally well written and addresses an intriguing and important topic that should be of general interest. The authors have revised the text to address my initial concerns, and I strongly recommend publication of this detailed and important work. Below I present some comments to improve the final version, but I think they should be simple to address.

R: Thank you very much for this concise summary and the positive comments on our work. We address your other comments in details below.

Comments:

2.1 In lines 569-570, the authors write: "This was mainly due to lower genome-wide transcription levels in females, including for autosomal genes..."

How the genome-wide transcriptional levels are being normalized is very important. Is this on a per animal basis, or a per cell basis? Are there basic facts about each sex and what size it reaches that might be relevant here? This section should be expanded to explain the critical background assumptions to the reader.

R: Thank you for raising this question. Since the transcriptome data used here are all derived from pooled whole individuals of either sex (for some species, individual worm is too small for preparing RNA-seq libraries), it reflects male or female population difference rather than per animal or per cell difference. During the revision, we looked into the individual numbers of each sample and also their sizes, most of the data that we analysed contained a similar number of males and females; and there are some sexual dimorphisms of sizes across species (Table 1). However, as far as we know, all previous analyses comparing gene expression levels between sexes did not take the individual size difference into account. We have now accordingly expanded the text at lines 483-496 to clarify this point.

Table 1. Sample and species information of transcriptome data used in this work.

Species	Male RNA-seq data	Female RNA-seq data	Morphology	Morphology references
Trichuris muris	mixed rear end of adult male worms, containing gonad tissue, from 35 adult male worms	mixed rear end of adult female worms, containing gonad tissue, from 65 adult female worms	Female larger. The females were 32 to 38 mm long, and the diameters measured 0.67 mm in the posterior and 0.1 mm in the anterior region. The males had a length of 26 to 30 mm and diameters of 0.48 mm in the posterior and 0.09 mm in the anterior region.	Wütemann M, Schmalz G, Mehlhorn H. Light and electron microscopic studies on two nematodes, Angiostrongylus cantonensis and Trichuris muris , differing in their mode of nutrition[J]. Parasitology research, 2007, 101(2): 225-232.
Trichuris suis	adult male whole body (don't know numbers)	adult female whole body	Female larger. The whipworm T. suis is a nematode commonly present in the caecum and colon of pigs. Adult worm size ranges from 3 to 8 cm with female being larger than male.	EFSA Panel on Nutrition, Novel Foods and Food Allergens (NDA), Turck D, Castenmiller J, et al. Safety of viable embryonated eggs of the whipworm Trichuris suis as a novel food pursuant to Regulation (EU) 2015/2283[J]. EFSA Journal, 2019, 17(8): e05777.
Strongyloides papillosus	free living adult males	free living adult females	Free-living females are slightly larger than males. Free-living female 770µm to 1.11mm, free-living male 700-825µm.	The morphology and development of the sheep nematode, Strongyloides papillosus (Wedl, 1856)
Strongyloides malayi	adult male whole bodies	adult female whole bodies	Female larger, female worms measure 43 to 55 mm in length by 130 to 170 µm in width, and males measure 13 to 23 mm in length by 70 to 80 µm in width.	https://www.cdc.gov/parasites/hyphalofarasia/biology_b_malay.html
Oncocerca volvulus	adult male whole body worms (2-5 worms)	adult female whole body worms (2-5 worms)	Female are much larger than male (10 folds longer), females measure 33 to 50 cm in length and 270 to 400 µm in diameter, while males measure 19 to 42 mm by 130 to 210 µm.	https://www.cdc.gov/nczod/dzdx/oncocerciasis/index.html
Haemonchus contortus	male adults (100-200 worms)	female adults (100-200 worms)	Female larger, the females have a length ranging from 18-30 mm, the males are shorter, ranging from 10-20 mm.	https://animaldiversity.org/accounts/Haemonchus_contortus/
Pristionchus pacificus	at least 750 young adult male worms	at least 750 young adult female worms	Female larger, the adult hermaphrodite P. pacificus is approximately 1 mm in length while the male is smaller and contains male sexual structures.	http://www.wormatlas.org/pristonchus/introduction/Pintrof_ramses.html

2.2 In lines 572-4, the authors write: "The transcription levels of autosomal genes in male *O. volvulus* are similar to those of their orthologs in *B. malayi*"

Here also it is critical to explain the method for comparing transcription levels between species, which is potentially complex.

R: We first extracted the 1-1 orthologous genes shared between *O. volvulus* and *B. malayi* using OrthoFinder v2.2.6, then calculated these genes' RPKMs for comparison. Only genes that are located on autosomes of both species are considered. We now added this part into the method, and labelled this as "see Methods" in the main text at line 493.

2.3 Line 45 would be a good place to add a paragraph break, to help the general reader consider each complex evolutionary transition separately. Furthermore, if the authors include a potential selective cause for the transition from hermaphroditism to androdioecy, they should also include one for the transition from gonochorism to androdioecy.

R: At line 50, we made a paragraph break and added "likely due to selection for reproductive assurance" after describing the transition from gonochorism to androdioecy. We also added a citation "Pannell JR. 2002. The Evolution and Maintenance of Androdioecy. Annual Review of Ecology and Systematics 33: 397-425."

2.4 In lines 686-7 the authors write: "Lineage-specific duplication of *C. elegans* sex determining genes across the phylogeny of nematodes" and in lines 708-11 they write: "Duplication of orthologs of genes involved in the sex-determination pathway of *C. elegans* also seems to have occurred in other nematodes, as we found lineage-specific duplications of sex-determination pathway genes on different Nigon elements"

Is it possible that some of these events are not lineage-specific duplications, but rather allelic variants in male/female populations? This concern should be addressed in the text.

R: It is not possible, as when we inferred the lineage-specific duplications, only the reference genome is used without involving male and female populations. And only genes and their duplicated copies that were annotated at different genomic locations, were considered as candidates for lineage-specific duplications.

2.5 In Figure 5k, the authors should revise the nematode portion of the model. (a) The initial part showing hermaphrodites evolving to make a male/female species is largely outside of the scope of this paper and it might be better to eliminate it so as to focus on a male/female species evolving into a male/hermaphrodite one. (b) In this second step, the *mss* genes are certainly very important, but *fog-2* and *she-1* should also be included, as they help control germ cell sex. In addition the authors should consider mentioning *try-5*, *spe-8* and other sperm activation genes.

R: We agree with (a) and revised the figure to show Nematode clade starting from a gonochoristic state, which is their ancestral state. However, we did provide some insights into the transition from a hermaphroditic state to a gonochoristic state by comparing the gonochoristic schistosomes to their hermaphroditic relatives (Fig. 5d-j). So we kept the hermaphroditic part of the schistosomes. We also added *fog-2*, and *she-1* onto the figure as suggested.

2.6 In line 684 the authors write: "the candidate sex-determining genes *U2AF2*, *fox-1*"
I can't see the arrow for *fox-1*

R: We revised the in paper that "the *fox-1* paralog of *S. mansoni* may not have a sex-determining function", so we removed the *fox-1* gene from the Fig. 5a and we also revised the legend text accordingly.

Minor comments

line 89 Change "an ancestral sex chromosomes." to "an ancestral sex chromosome."

R: Revised (see line 94-95).

line 134 Change "for future discovering the sex determining" to "for the future discovery of the sex-determining"

R: Revised (see line 138).

line 164 Change “has also previously been used to detected” to “has also previously been used to detect”

R: Revised (see line 167-168).

line 187 Change “and presence” to “and the presence”

R: Revised (see line 187).

line 204 Change “due to presence” to “due to the presence”

R: Revised (see line 199).

line 264 Change “creating a neo-sex chromosomes” to “creating a neo-sex chromosome”

R: Revised (see line 241).

line 334 Change “for a long evolutionary time” to “for such a long evolutionary time”

R: Revised (see line 308).

line 366 Change “elements that forms the X” to “elements that form the X”

R: We’ve deleted this sentence.

line 403 Change “specifically in male” to “specifically in males”

R: Revised (see line 336-337).

line 576 Fix “as suggested in Ref.”

R: Fixed.

line 648 Change “in the focal species in future.” to “in the focal species in the future.”

R: Revised (see line 550-551).

line 648 Change “our description of” to “our description on”

R: Revised (see line 551).

line 669 Change “as the first step of evolution” to “as the first step in the evolution”

R: Revised (see line 571).

line 701 Change “model that for” to “model for”

R: Revised (see line 1276).

Reference

Post R. 2005. The chromosomes of the Filariae. *Filaria J* 4: 10.